# POST ✉: A Framework for Privacy of Soft-prompt Transfer

## Abstract

Prompting has emerged as a dominant learning paradigm for adapting large language models (LLMs). While discrete (textual) prompts prepend tokens to the input for optimized outputs, soft (parameter) prompts are tuned in the embedding space via backpropagation, requiring less engineering effort. However, unlike semantically meaningful discrete prompts, soft prompts are tightly coupled to the LLM they were tuned on, hindering their generalization to other LLMs. This limitation is particularly problematic when efficiency and privacy are concerns, since (1) it requires tuning new prompts for each LLM which, due to the backpropagation, becomes increasingly computationally expensive as LLMs grow in size, and (2) when the LLM is centrally hosted, it requires sharing private data for soft prompt tuning with the LLM provider. To address these concerns, we propose a framework for Privacy Of Soft-prompt Transfer (POST), a novel method that enables private soft prompt tuning on a small language model and then transfers the prompt to the large LLM. Using knowledge distillation, we first derive the small language model directly from the LLM to facilitate prompt transferability. Then, we tune the soft prompt locally, if required with privacy guarantees, *e.g.*, through differential privacy. Finally, we use a small set of public data to transfer the prompt from the small model to the large LLM without additional privacy leakage. Our experimental results demonstrate that our method effectively transfers soft prompts while protecting client data privacy while also reducing the computational complexity compared to soft prompt tuning on the large model.

## 1 Introduction

Large Language Models (LLMs) are strong general purpose language generators that can be adapted to solve various private downstream tasks (OpenAI, 2023; Gemini-Team et al., 2023). One prominent paradigm for adapting LLMs to private tasks is prompting (Devlin et al., 2018; Radford et al., 2018). While *discrete prompts* (Schick & Schütze, 2020; 2021a; Shin et al., 2020; Han et al., 2022) which prepend textual tokens to the LLM's input have been shown relatively successful for LLM adaptations, they require large engineering efforts and lots of trials and errors. As an alternative, *soft prompts* (Shin et al., 2020; Lester et al., 2021; Li & Liang, 2021; Zhong et al., 2021; Oymak et al., 2023) prepend trainable embedding vectors to the input, which can be tuned automatically on the private downstream data using standard gradient-based approaches. Such gradient-based approaches are generally known to yield higher performance at lower computational costs (Liu et al., 2022).

Yet, soft prompt tuning has two major limitations. 1) As LLMs grow in size (Geng & Liu, 2023; Chiang et al., 2023; Brown et al., 2020), it requires significantly more computation to backpropagate through the entire LLM. 2) Backpropagation requires the model and data to be on the same device. In the current model of centrally hosted LLMs, this requires users to share their data with the LLM provider, which may be of concern when this data is private or sensitive in nature. Alternatively, the LLM provider could share their model with the client, mitigating user data privacy concerns. However, this would put the LLM provider's intellectual property at risk and disrupt their business model, as users would no longer be required to pay per model query. Further, the computational resources required to backpropagate through the model would make this impractical for most clients.

A potential solution to both problems is to tune the soft prompt locally on a smaller model and then transfer and use it on the large LLM. This approach, commonly known as "prompt transfer" (Su

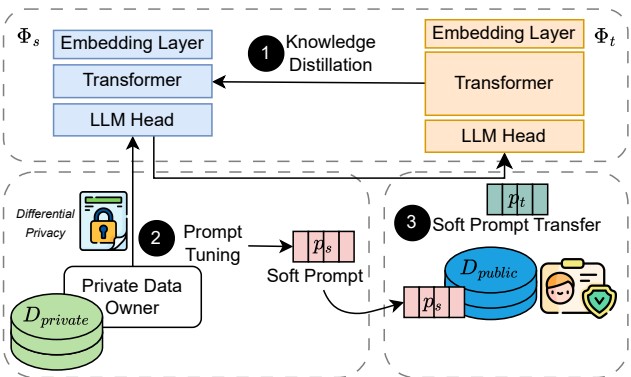

Figure 1: **POST✉ Framework.** ❶ An LLM provider compresses their LLM $\Phi_t$ into a smaller language model $\Phi_s$ by using knowledge distillation. ❷ The private data owner learns a specific soft prompt $p_s$ on $\Phi_s$ using their private dataset (optionally with differential privacy guarantees). ❸ The LLM provider obtains the soft prompt $p_t$ for solving the user's task by transferring $p_s$ to the target LLM $\Phi_t$—solely relying on a small public dataset and no access to the private data for transfer.

et al., 2022; Wu et al., 2023b; Xiao et al., 2023), has proven effective for discrete prompts that carry semantic meaning (Rakotonirina et al., 2023; Hong et al., 2023; Wen et al., 2023). However, soft prompts are highly coupled to the LLM they were tuned on, making them difficult to transfer. Existing approaches for transferring soft prompts between two LLMs have one of two issues: either they require private data access for central large model (Su et al., 2022), which as we discuss is not feasible when data privacy is of concern, or they are ineffective, as the transferred prompt's utility on the large central LLM often underperforms compared to the prompted small model (Wu et al., 2023b), disincentivizing the use of the large model altogether.

To address these challenges, we propose **POST**, a framework for **P**rivacy **O**f **S**oft-prompt **T**ransfer. POST consists of three key steps. (1) The LLM provider performs *knowledge distillation* (Hinton et al., 2015) to compress their LLM into a smaller language model. This smaller model is designed to meet three critical requirements: it must (i) be small enough to enable the user to perform local soft prompt tuning on their own hardware, (ii) closely match the semantics of the original LLM to facilitate an effective prompt transfer, and, from the perspective of the LLM provider, (iii) be limited in performance to ensure that users still have an incentive to use the original LLM rather than the small prompted version. The LLM provider sends the distilled small model to the user. Then, (2) the user then performs *local prompt tuning* using their private data on this smaller model, potentially incorporating formal privacy guarantees through differential privacy (Dwork et al., 2006). Once the user has tuned the private prompt on their data, they provide this prompt to the LLM provider, who then (3) *transfers the prompt* to achieve strong performance on the large LLM. To prevent any privacy leakage from the user's private data, we equip POST with a novel prompt transfer method that relies purely on access to a small public dataset rather than the user's private data for transfer.

Our thorough experimental evaluation on both masked language models and auto-regressive language models demonstrates that our method can efficiently, effectively, and privately transfer soft prompts with high utility. In summary, we make the following contributions:

- We propose POST, a framework for privacy of soft prompt transfer. POST preserves confidentiality of users' private data and can also provide strong privacy guarantees through differential privacy.

- We design a novel method to transfer private prompts between LLMs by purely relying on public data which we integrate into POST.

- We provide detailed experimental analysis using four public datasets to simulate our setup and two different types of LLMs to show the effectiveness and efficiency of our method.

## 2 BACKGROUND

**Prompt Tuning.** Prompt tuning (PT) aims at adapting a publicly pre-trained LLM to various natural language downstream tasks. There are two major types of prompts, 1) *hard or discrete prompts* (Schick & Schütze, 2021a;b; Gao et al., 2021), which are discrete textual tokens prepended to the input text of the LLM, and 2) *soft prompts* (Hambardzumyan et al., 2021; Qin & Eisner, 2021; Zhong et al., 2021) which are tunable embedding vectors prepended to the LLM's input. While discrete prompts require thorough engineering to yield good performance on downstream tasks, soft prompts can be tuned through standard gradient-based training approaches (Lester et al., 2021). Formally, given an input sequence with $n$ tokens $X = \{x_1, x_2, \ldots, x_n\}$, labeled by $y$, we fist prepend $l$ randomly initialized soft prompts $P = \{\mathbf{p}_1, \mathbf{p}_2, \ldots, \mathbf{p}_l\}$ before $X$, where $\mathbf{p}_i \in \mathbb{R}^d$ is an embedding vector, and $d$ is the input dimension of the LLM,. The training objective is to maximize the likelihood of decoding the correct output $y$ as $\mathcal{L} = p(y|P, X)$. The key is that the model itself remains frozen and only $P$ is tunable.

**Knowledge Distillation.** Knowledge Distillation (KD) (Hinton et al., 2015; Buciluǎ et al., 2006) is a compression method for machine learning models. It works by transferring knowledge from a complex model, denoted as the *teacher model*, to a simpler smaller model known as the *student model*. KD has been shown effective to compress LLMs during the pre-training phase while maintaining their performance (Sanh et al., 2019; Gu et al., 2024; Sreenivas et al., 2024). Already pre-trained LLMs can also be compressed successfully through KD (Gu et al., 2024). In prompt transfer, Zhong et al. (2024) leverage knowledge distillation to alleviate knowledge-forgetting between tasks that use transferred prompts. In contrast, our setup considers transferring prompts between models. While, in general, most of the KD approaches aim at generating a student with a similar predictive performance as the teacher, for our purpose, the student performance is not particularly relevant. The student just needs to match the teacher's predictive behavior to a certain degree in order to facilitate the transfer of the prompt from student to teacher.

**Soft Prompt Transfer.** Since soft prompts are trained with backpropagation through the LLMs, this process can be computationally expensive, especially as LLMs grow in size. This motivates the emergence of attempts to transfer, *i.e.,* to reuse, existing soft prompts. There are two broad scenarios for prompt transfer. The first one aims at reusing a soft prompt trained for one downstream task on another (similar) downstream task on the same LLM (*cross-task transfer)*. This can be implemented, for example, by initializing the parameters of the second soft prompt with the trained existing soft prompt parameters and has been shown to reduce training time for the second prompt (Vu et al., 2022; Su et al., 2022; Zhong et al., 2024). The second and more challenging scenario for prompt transfer is a *cross-model transfer* scenario. In this scenario, one tries to tune a prompt for a given task on one LLM, and then use it for another LLM. The difficulty arises from the fact that soft prompts (over)fit the LLM they were tuned for and usually do not exhibit a strong performance on other LLMs. Su et al. (2022) address transferring the soft prompt between the LLMs by using the guidance of the private data. However, this approach still exposes the data directly to the second LLM which may not be possible when this LLM is hosted centrally by a service provider (e.g., OpenAI) and the data is sensitive in nature, as the private data now needs to be shared with the external provider. Wu et al. (2023b) present a zero-shot prompt transfer method, where source prompts tuned on a given LLM are encoded into a relative space and used as a form of support vector when finding target prompts on the second, *i.e.,* target model. Unfortunately, in their approach, the target model with the transferred prompt performs worse than the prompted source model, leaving no incentive to use the target model rather than the source model with the prompt. In contrast, our method significantly improves performance on the target models with the transferred prompts. Additionally, their transfer requires the private data, which thereby, leaks entirely to the model owner. In contrast, our method performs prompt transfer with public data, preserving confidentiality of the private data towards the model provider. Transferring tasks between LLM has also been explored by Xiao et al. (2023) for transfer learning. While they focus on fine-tuning and their approach is not applicable for soft-prompt tuning, we operate in the same setup and under the same assumptions as they are.

**Differential Privacy for Soft Prompts.** Differential privacy (DP) (Dwork, 2006) is a mathematical framework that provides privacy guarantees for ML by implementing the intuition that a model $\mathcal{M} : I \rightarrow S$, trained on two neighboring datasets $D$, $D'$ that differ in only one data point, will yield

roughly the same output, *i.e.,* $\Pr[\mathcal{M}(D) \in S] \leq e^{\epsilon} \cdot \Pr[\mathcal{M}(D') \in S] + \delta$. The privacy parameter $\varepsilon$ specifies by how much the output is allowed to differ and $\delta$ is the probability of failure to meet that guarantee. To adapt soft prompts with DP guarantees, Duan et al. (2024) proposed the PromptDPSGD algorithm, which is based on the popular differentially private stochastic gradient descent algorithm (DPSGD) (Abadi et al., 2016). To obtain a finite DP guarantee, each gradient must be clipped and calibrated Gaussian noise added to the sum.

**Private Prompting and Text-to-Text Privatization.**   There exist multiple DP frameworks for private prompting (Duan et al., 2023; Tang et al., 2024; Wu et al., 2024). However, they mainly operate in a different setup and only provide DP guarantees for the model output, yet leak the private data to the model provider. In contrast, our work aims at protecting the private data against the model provider. In a similar vein to our work, **DP-OPT** (Hong et al., 2023) tries to avoid leakage to the model owner and tunes discrete prompts with DP guarantees on a local surrogate LLM and then transfers these prompts to the large LLM. Their focus on discrete prompts (in contrast to our work that relies on soft prompt) leads to certain words and phrases from the training dataset leaking directly to the LLM provider, as shown in their Figure 3—which does not occur with our method. Additionally, their results show that the local surrogate model needs to be of strong performance (*i.e.,* large) to obtain a good transfer results, leading to very high compute requirements on the user side. In contrary to our method relies on small surrogate models that can be used for prompt tuning with low compute on the user's end.

## 3   SETUP AND PROBLEM FORMULATION

**The Setup.**   We consider two parties: an LLM provider and a user, as shown in Figure 2. The LLM provider deploys a general-purpose LLM and offers paid query access to it. The user holds private data and wants to adapt the LLM on this data to solve their downstream tasks while ensuring the confidentiality and privacy of their data towards the LLM provider.

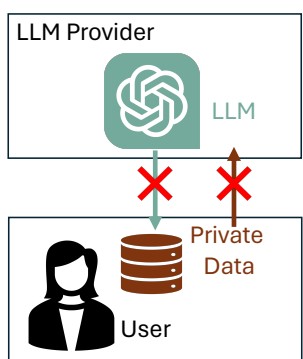

Figure 2: **The Setup.**

**The Problem.**   Unfortunately, both the data and the LLM are required to be on the same device to faciliate the computation of gradients of the model's predictions on the private data with respect to the soft prompt. The problem is that the user may not be able to share their data with the LLM provider due to privacy concerns while the LLM provider cannot share their LLM because of 1) intellectual property concerns and since 2) this would disrupt their business model, as users would no longer be required to pay for accessing model queries. Additionally, most users would lack the necessary computational resources to tune the soft prompt on the large LLM locally, as this requires calculating gradients over the entire model. Due to these limitations, the powerful LLM cannot be used for private tasks.

**Our Solution.**   We propose a solution based on tuning the soft prompt on a small local model and then transferring this prompt to the LLM by using public data. To obtain a suitable small model that facilitates prompt transfer, we propose that the LLM provider performs KD from their LLM. The resulting small model should be (i) small enough such that the user can tune it on local hardware. At the same time, (ii) it should, to a certain degree, match the predictive behavior of the large one to facilitate the transfer of the prompt. However, it should (iii) not exhibit too high generalization performance, as the user might otherwise just tune the prompt on that model and use it for their downstream task without paying access to the large model to the LLM provider. After distillation, the small model is sent to the user who tunes a soft prompt on it using their private data, potentially with DP to formally bound privacy leakage. Finally, the tuned prompt is sent to the LLM provider who performs a prompt transfer step for the private prompt relying on public data. Then, the client can use the LLM using the transferred prompt. We provide an overview of this solution in Figure 1 and detail its building blocks in the following section.

# 4 OUR PRIVATE TRANSFER OF SOFT PROMPTS FRAMEWORK

Our **P**rivacy **O**f **S**oft-prompt **T**ransfer (POST) framework consists of three main building blocks, (1) a knowledge distillation from the LLM to a small model, (2) private prompt tuning, and (3) a privacy-preserving prompt transfer using public data. We detail those building blocks below.

## 4.1 KNOWLEDGE DISTILLATION

We denote the teacher LLM model as $\Phi_t$ and the small student model as $\Phi_s$. The input sequence to an LLM is denoted as $x$. We leverage KD in (Sanh et al., 2019) to derive $\Phi_s$ from $\Phi_t$. Different from previous work in LLM distillation (Sanh et al., 2019; Xiao et al., 2023) that moderately compresses the LLM and tunes the whole model to recover performance as much as possible, we perform a more aggressive KD without emphasis on the student model's performance. In detail, we rely on the following loss from (Sanh et al., 2019) to distill $\Phi_s$ from $\Phi_t$:

$$\mathcal{L}_{distil} = \alpha_{ce}\mathcal{L}_{ce} + \alpha_{lm}\mathcal{L}_{lm} + \alpha_{cos}\mathcal{L}_{cos}. \tag{1}$$

The objective is a linear combination of distillation loss $\mathcal{L}_{ce}$, language modeling loss $\mathcal{L}_{lm}$ and embedding cosine loss $\mathcal{L}_{cos}$. Where $\mathcal{L}_{ce}$ is the Kullback–Leibler divergence loss between the logits of $\Phi_s$ and $\Phi_t$, $\mathcal{L}_{lm}$ is the standard language modelling pretrainign objective, *i.e.,* the cross entropy loss for predicting the masked/next tokens, and $\mathcal{L}_{cos}$ is the cosine distance of the embedding of $\Phi_s$ and $\Phi_t$ with $\alpha_{ce}, \alpha_{lm}$ and $\alpha_{cos}$ weighting the respective losses.

Building on our intuition that more similar models exhibit better prompt transfer, we assess different ways of preserving this similarity during KD. We observe that fixing the language modeling head, *i.e.,* causing higher output similarity, leads to slightly better transfer performance. Thus we use this strategy inside our KD. In contrast, we did not observe a consistent improvement with fixing the embeddings. Our ablation studies are shown in Appendix E.1 and the final detailed distillation setup is presented in Appendix C.1.

## 4.2 PRIVATE PROMPT TUNING

The goal is to tune a local prompt $p_s$ on the small source model $\Phi_s$ using the private data $D_{pri}$ such that $p_s$ minimizes the loss $\mathcal{L}$ on the private downstream task as

$$\arg\min_{p_s} \sum_{x \in D_{pri}} \mathcal{L}(\Phi_s, p_s + x). \tag{2}$$

This approach can be performed with standard PT. However, this only provides confidentiality for the private data since the data is not directly sent to the LLM provider. Recent work (Duan et al., 2023), however, highlights that private information can leak from tuned prompts.

To formally bound privacy leakage, $p_s$ can also be tuned with DP guarantees, for example, using the PromptDPSGD algorithm (Duan et al., 2024). During optimization, PromptDPSGD clips the per-sample gradients of the loss to a clip norm $c$ and adds Gaussian noise drawn from $\mathcal{N}(0, \sigma^2 c^2)$ to provide $(\varepsilon, \delta)$-DP guarantees.

## 4.3 PRIVACY-PRESERVING PROMPT TRANSFER THROUGH PUBLIC DATA

The prompt $p_s$, tuned on the small source model $\Phi_s$, could, in principle, be directly applied to the large target LLM $\Phi_t$. However, as described above, soft prompts fit very strongly the model that they were tuned on. Hence, they do not initially perform very well on other LLMs. A naive solution is to fine-tune the target prompt $p_t$ on the private data $D_{pri}$. However, this would disclose the private data to the LLM provider and is, hence, not acceptable. As an alternative, we propose a privacy-preserving prompt transfer that leverages a small public data $D_{pub}$ in an efficient transfer step to derive a high-utility prompt $p_t$ from $p_s$.

We start by initializing the target prompt $p_t$ with the same initialization of $p_s$, then iteratively update $p_t$. For the iterative update, we use the loss function

$$\mathcal{L} = (1 - \alpha)\mathcal{L}_1 + \alpha\mathcal{L}_2, \tag{3}$$

that consists of two different loss terms. The first loss term is defined as

$$\mathcal{L}_1 = \sum_{\hat{x} \in D_{pub}} \text{KLDiv}(\Phi_t(p_t + \hat{x}), \Phi_s(p_s + \hat{x})), \tag{4}$$

where KLDiv denotes the Kullback–Leibler divergence. It aims for aligning the predictions of the prompted source and target model on the public data. The second loss term is defined by

$$\mathcal{L}_2 = \sum_{\hat{x} \in D_{pub}} \text{KLDiv}((\Phi_t(p_t + \hat{x}) - \Phi_t(\hat{x})), (\Phi_s(p_s + \hat{x}) - \Phi_s(\hat{x})), \tag{5}$$

and optimizes to align the direction change induced by the private prompt between $\Phi_t$ and $\Phi_s$, again on the public data.

The hyperparameter $\alpha$ in Equation (3) controls the balance between the two loss terms. We observe that a good choice of $\alpha$ depends largely on the model's zero-shot performance. We tend to use larger $\alpha$ when the teacher LLM $\Phi_t$ already has a non-trivial zero-shot performance on the private task. The intuition is that if the model performs well, it needs to less mimic the behavior of the smaller model, but only incorporate that model's direction change induced by the prompt. On the other side, when $\Phi_t$ has poor zero-shot performance, we put more emphasis on the output of the compressed model to provide the update. In Table 10, we present the $\alpha$ values chosen in our experiments. Additionally, we conduct an ablation study with those $\alpha$ in Appendix E.3. The ablation shows that while our method is robust to the choice of $\alpha$, includig both loss terms outperforms just using one, highlighting the necessity of our design.

## 5 EMPIRICAL EVALUATION

### 5.1 EXPERIMENTAL SETUP

**Models and Datasets.** To obtain the compressed model, we follow Sanh et al. (2019) to aggressively distill a 12-layer Roberta-base (Liu et al., 2019) into a 2-layer model and a 48-layer GPT2-XL (Radford et al., 2019) into a 4-layer model, and a 32-layer Llama2-7b (Touvron et al., 2023) into a 2-layer small model. We use the Bookcorpus (Zhu et al., 2015) dataset for distillation.

We evaluate the performance of our proposed method on four classification datasets: sst2 from the GLUE benchmark (Wang et al., 2019), imdb (Maas et al., 2011), tweet (Rosenthal et al., 2017) and arisetv (chimaobi Samuel, 2022). We use these four datasets to simulate private and public data by selecting two different datasets, one as private data and one as public data.[1] When choosing the public dataset, we also include agnew (Zhang et al., 2015). We discuss the choice of the public datasets for transfer in more detail in Appendix C.4. We follow Li et al. (2022) to formulate the classification task as a text-infilling task. *e.g.,* for masked language models such as Roberta, we append "it was ¡mask¿" to the input and let the model predict the ground truth text. The setting for GPTs is similar in that we append "it was" to predict the next word. We report the ground truth text used in our experiments in Appendix C.2.

**KD, Prompt Tuning, and Prompt Transfer.** We follow (Sanh et al., 2019) to set the hyperparameters of knowledge distillation (see Appendix C.1 for details). To train soft prompt, we follow settings in Su et al. (2022). When applying DP, we use PromptDPSGD (Duan et al., 2024). Prompt tuning settings are presented in Appendix C.3. During the prompt transfer, the model provider has no access to the private dataset to find the right moment to stop the transfer, so we report the transferred accuracy at fixed steps. We use 5000 steps for Roberta-base and 8000 steps for GPT2-XL. For each private dataset, we report the transfer performance obtained using two different public datasets. We also conduct experiments with varying steps and the number of public data points used for transfer.

**Metrics and Baselines.** To evaluate the success of our method, we report the accuracy on the test data split of our private datasets for the teacher LLM with the transferred prompt (**Private Transfer**). As baselines for comparison, we include the zero-shot performance of the teacher LLM on the private tasks' test sets (**Full ZS**), representing the lower bound our method should improve upon. Additionally, we provide the performance of tuning the prompt for the teacher LLM on the private training data, which, due to privacy concerns, is not feasible in practice (**Full PT**). This serves as

---

[1]Note, we use public datasets to simulate private data.

Table 1: **Confidential prompt transfer performance.** We compress Roberta-base, GPT2-XL and Llama2-7b, tune prompts for different private dataset on the compressed models, and transfer them back using different public datasets (POST). As baselines, we present the large models' zero-shot performance on the private data (Full ZS), the accuracy of tuning the prompt with the private data on the large models (Full PT) and the small model (Compressed PT), and the performance of the prompt tuned on the small model when direcly applied to the large one (Direct Transfer). Our POST significantly improves performance over the small prompted model and our prompt transfer yields a strong improvement over the direct transfer.

|        |         |         |               |                 | POST (ours) |          |        |          |
| Private | Full ZS | Full PT | Compressed PT | Direct Transfer | Public | Test acc | Public | Test acc |
| --- | --- | --- | --- | --- | --- | --- | --- | --- |
| sst2    | 72.25 | 91.74 | 79.10 | 76.49 | tweet | **87.73** | imdb | 85.21 |
| imdb    | 72.19 | 89.88 | 78.85 | 76.92 | tweet | **83.96** | sst2 | 80.27 |
| tweet   | 36.53 | 68.68 | 56.65 | 43.10 | imdb  | 54.55 | sst2 | **58.25** |
| arisetv | 38.80 | 89.81 | 70.98 | 47.82 | agnews | **82.73** | tweet | 68.48 |

(a) Roberta-base.

|        |         |         |               |                 | POST (ours) |          |        |          |
| Private | Full ZS | Full PT | Compressed PT | Direct Transfer | Public | Test acc | Public | Test acc |
| --- | --- | --- | --- | --- | --- | --- | --- | --- |
| sst2    | 60.78 | 94.84 | 80.94 | 59.06 | tweet | **85.89** | imdb | 83.49 |
| imdb    | 60.27 | 93.28 | 81.32 | 60.34 | tweet | **83.93** | sst2 | 82.15 |
| tweet   | 34.71 | 68.60 | **63.13** | 41.50 | imdb  | 61.75 | sst2 | 57.70 |
| arisetv | 52.98 | 92.45 | 77.10 | 55.43 | agnews | **87.56** | tweet | 82.12 |

(b) GPT2-XL.

|        |         |         |               |                 | POST (ours) |          |        |          |
| Private | Full ZS | Full PT | Compressed PT | Direct Transfer | Public | Test acc | Public | Test acc |
| --- | --- | --- | --- | --- | --- | --- | --- | --- |
| sst2    | 78.67 | 94.84 | 78.78 | 55.28 | tweet | 89.33 | imdb | **90.14** |
| imdb    | 83.74 | 97.02 | 79.95 | 70.57 | tweet | **86.27** | sst2 | 86.25 |
| tweet   | 44.50 | 72.03 | 54.12 | 41.70 | imdb  | 57.55 | sst2 | **61.70** |
| arisetv | 76.57 | 93.47 | 77.92 | 54.23 | agnews | **86.71** | tweet | 79.59 |

(c) Llama2-7b.

the theoretical upper bound for potential performance. We also report the accuracy of the prompted compressed model after tuning the prompt on it (**Compressed PT**), as our private transfer must improve over this metric to justify using the teacher LLM instead of the small prompted one. Finally, we report the direct transfer accuracy (**Direct Transfer**), which is the accuracy achieved when the prompt tuned on the small model is directly applied to the large one, highlighting the effectiveness of our prompt transfer step.

## 5.2 PRIVATE PROMPT TRANSFER WITH POST

**Confidential Transfer.** In Table 1, we evaluate the performance of our method in a scenario where only the confidentiality of the private data is protected. Therefore, the user locally tunes a soft prompt without DP guarantees. For each private dataset, we experiment with two different public datasets for prompt transfer and report the respective transferred accuracy on the private dataset. We first observe that the transferred performance is significantly higher than the zero-shot performance. Additionally, after prompt transfer with POST, we outperform the small compressed prompted model, giving users a strong incentive to transfer their prompt back to the teacher LLM. Further, we show that our prompt transfer described in Section 4.3 is highly effective as it improves over the direct transfer performance by a large margin. We do observe that the choice in the public dataset can sometimes have an impact on the final test-performance, which can be resolved with additional tuning. In contrast to the soft prompt transfer method by Wu et al. (2023b) which showed a *decrease* in accuracy after transfer, our results highlight the practical applicability and the benefits of using our method.

**Differntially Private and Confidential Transfer.** In addition to providing confidentiality, POST is easily amenable to providing provable privacy guarantees through DP, which protects against the LLM provider and third parties who may observe the tuned prompt and the model's outputs on it. Here, we tune the local prompt with DP. Since the prompt transfer is executed using a few *public* data points, no additional privacy leakage is incurred in that step. We show the results of our experiments with privacy guarantees for $\varepsilon = 8$ in Table 2. The trends observed for the confidential prompt transfer also hold under local soft prompt tuning with DP. In particular, we observe that the improvement of the transfer performance to the large LLM over the performance on the prompted compressed

Table 2: **Differentially Private and Confidential prompt transfer performance.** We compress Roberta-base, GPT2-XL and Llama2-7b, tune prompts for different private dataset on the compressed models with Differential Privacy guarantees ($\varepsilon = 8$), and transfer them back using different public datasets (POST). As baselines, we present the large models' zero-shot performance on the private data (Full ZS), the accuracy of PromptDPSGD tuned prompt with the private data on the large models (Full PT) and the small model (Compressed PT), and the performance of the prompt tuned on the small model when direcly applied to the large one (Direct Transfer). Our POST significantly improves performance over the small prompted model and our prompt transfer yields a strong improvement over the direct transfer.

| Private | Full ZS | Full PT | Compressed PT | Direct Transfer | POST (ours) | | | |
| | | | | | Public | Test acc | Public | Test acc |
|---|---|---|---|---|---|---|---|---|
| sst2 | 72.25 | 90.14 | 67.54 | 77.06 | tweet | **84.40** | imdb | 81.42 |
| imdb | 72.19 | 88.55 | 72.22 | 74.35 | tweet | 79.64 | sst2 | **80.64** |
| tweet | 36.53 | 62.05 | 40.87 | 43.15 | imdb | 55.65 | sst2 | **59.25** |
| arisetv | 38.80 | 80.33 | 64.25 | 47.34 | agnews | **79.11** | tweet | 71.98 |

(a) Roberta-base.

| Private | Full ZS | Full PT | Compressed PT | Direct Transfer | POST (ours) | | | |
| | | | | | Public | Test acc | Public | Test acc |
|---|---|---|---|---|---|---|---|---|
| sst2 | 60.78 | 91.28 | 74.31 | 57.80 | tweet | 79.93 | imdb | **84.06** |
| imdb | 60.27 | 89.59 | 74.81 | 63.66 | tweet | **78.03** | sst2 | 75.16 |
| tweet | 34.71 | 61.47 | 48.60 | 41.50 | imdb | **58.05** | sst2 | 54.75 |
| arisetv | 52.98 | 83.24 | 67.16 | 57.25 | agnews | **82.12** | tweet | 80.55 |

(b) GPT2-XL.

| Private | Full ZS | Full PT | Compressed PT | Direct Transfer | POST (ours) | | | |
| | | | | | Public | Test acc | Public | Test acc |
|---|---|---|---|---|---|---|---|---|
| sst2 | 78.67 | 90.60 | 70.99 | 53.55 | tweet | 87.50 | imdb | **89.91** |
| imdb | 83.74 | 91.47 | 70.26 | 68.61 | tweet | 82.14 | sst2 | **83.26** |
| tweet | 44.50 | 62.40 | 48.16 | 41.65 | imdb | 56.60 | sst2 | **59.55** |
| arisetv | 76.57 | 83.73 | 64.43 | 64.73 | agnews | **82.60** | tweet | 75.24 |

(c) Llama2-7b.

model is even more significant than in the non-DP setup. For example, on the sst2 dataset, using tweet as public data, for Roberta-base, we observe an improvement of 16.86% for the DP case, while we only have an improvement of 8.63% in the non-DP case (see first lines of Table 1 and Table 2, respectively). We hypothesize that the noise added for DP during tuning acts as a regularizer that can help to prevent overfitting on the small sensitive datasets and the distilled model, hence, generalizing better to the large LLM.

## 5.3 EFFECT OF NUMBER OF PUBLIC SAMPLES USED FOR TRANSFER

We also investigate the influence of the size of the public dataset required to complete the transfer. Our results in Figure 3 show that we can already yield high transfer performance with less than 100 public data points. This small size of public datasets needed makes our method highly practical.

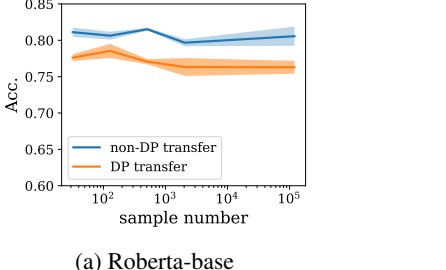

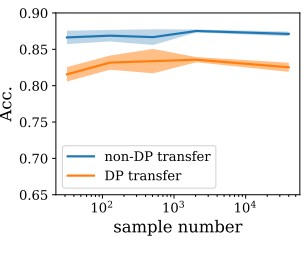

(a) Roberta-base

(b) GPT2-XL

Figure 3: **Effect of number of public samples.** We depict the number of samples from the public dataset used to perform our prompt transfer. We plot results for arisetv as the private dataset with data subsampled from agnews as public data. Our results highlight that with even less than 100 public data samples, our transfer yields high performance.

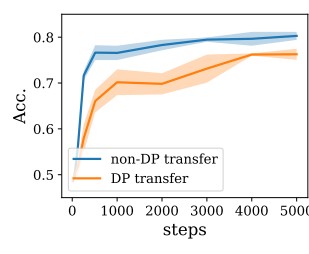 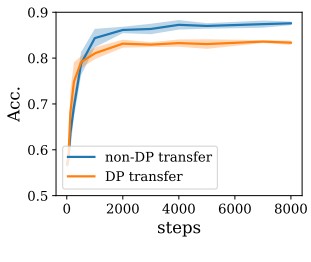

(a) Roberta-base  (b) GPT2-XL

Figure 4: **Effect of number of transfer steps.** We vary the number of steps during our private prompt transfer. We plot results for arisetv as the private dataset and agnews as public data. We observe that already a small number of transfer steps yields high performance.

Table 3: **Runtime of POST vs. Full PT.** We present the runtime for our method, split by its individual components and compare against full prompt tuning on the large LLM. We use arisetv and sst2 as private data. We execute 5000 steps of transfer. PT on $\Phi_t$, $\Phi_s$ takes 20 epochs until convergence. All experiments are executed on a single A100 GPU.

| Method | Runtime for arisetv (min) | Runtime for sst2 (min) |
|---|---|---|
| PT on $\Phi_t$ | 184 | 2660 |
| (1) PT on $\Phi_s$ | 23 | 310 |
| (2) Transfer | 99 | 99 |
| **Ours total (1)+(2)** | **122** | **409** |

## 5.4 Effect of Number of Transfer Steps

We additionally investigate how many transfer steps are required to obtain good performance. Based on the insights from the previous section, we randomly subsample 128 samples from the agnews dataset as public data and report the achieved accuracy on arisetv as private data over different numbers of transfer steps. Our results in Figure 4 highlight that only a small number of transfer steps is enough for convergence and high accuracy on the private task. We observe convergence within around 2000 steps for GPT2-XL and aroudn 1000 for Roberta-base/

## 5.5 Runtime of our Method vs. Full Prompt Tuning on the Large Model

While, in practice, tuning the large LLM with the private data can exhibit severe privacy risks and is, hence, not applicable, we compare runtimes to get an insight on the computational gains introduced by tuning the prompt on a small model and then transferring it. Since the PT time is determined by the size of the dataset if we want to backpropagate over all private training examples, we present the runtimes of our approach vs. prompt tuning on the large LLM for two different-sized datasets in Table 3. While on the small arisetv dataset, PT on the large model takes 150% of the time of executing our POST, for the larger sst2 datasets, our method improves the runtime roughly by a factor of six (409 instead of 2660 minutes on an A100). These results highlight that beyond the privacy protection, our POST also yields substantial improvements in computational efficiency.

## 5.6 Analyzing Privacy Leakage from the Soft Prompt based on MIA

We further analyze the risk of potential membership inference attacks (MIAs) (Shokri et al., 2017) against our locally tuned soft prompt. In the context of prompt tuning, these attacks try to identify whether a given data point was used to tune a given prompt (Duan et al., 2023; Wu et al., 2023a). We use a threshold-based membership inference attack and compare the prediction probabilities for members (private data used to tune the soft prompt) and non-members (private data not used to tune the soft prompt). In both cases, the output probabilities for members and non-members are (nearly) indistinguishable, demonstrating that MIA is ineffective in our scenario, as depicted in Figure 5. We hypothesize that this ineffectiveness stems from the small number of parameters being tuned for the soft prompt on the private dataset. DP adds additional protection by aligning the two distributions

Table 4: **Baseline comparison.** We present the performance of our method against state-of-the-art baselines. We report test accuracies over different private datasets $D_{pri}$. For our POST, we report the accuracies under the best public dataset (see Table 1 and Table 2).

| Method | $\Phi_t$ | $\Phi_s$ | sst2 | imdb | tweet | arisetv |
|---|---|---|---|---|---|---|
| OPT (Hong et al., 2023) | Llama2-7b | our compressed | 81.31 | 67.40 | 26.90 | 82.00 |
| OPT (Hong et al., 2023) | Llama2-7b | GPT2 | 81.65 | 62.93 | 41.15 | 78.26 |
| Zero-Shot Transfer (Wu et al., 2023b) | Llama2-7b | our compressed | 62.38 | 70.57 | 42.80 | 58.33 |
| Zero-Shot Transfer (Wu et al., 2023b) with DP | Llama2-7b | our compressed | 53.55 | 69.47 | 41.65 | 59,54 |
| **POST (ours)** | Llama2-7b | our compressed | **90.14** | **86.27** | **61.70** | **86.71** |
| **DP-POST (ours)** | Llama2-7b | our compressed | 89.91 | 83.26 | 59.55 | 82.60 |

even more. The ineffectiveness of the attack can be explained by the fact that our distilled model is too small to yield enough memorization that would enable telling members and non-members apart.

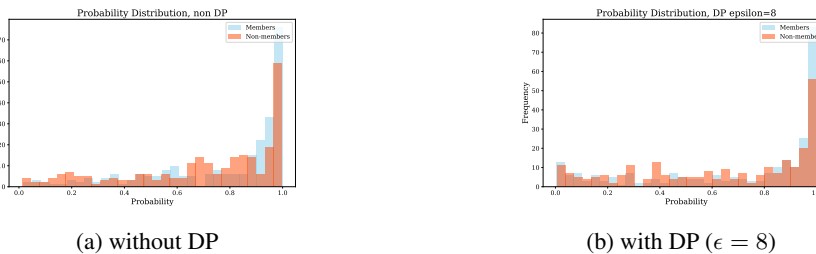

(a) without DP                    (b) with DP ($\epsilon = 8$)

Figure 5: **MIA risk of prompt trained on distilled Roberta-base without and with DP (sst2).**

### 5.7    COMPARING AGAINST STATE-OF-THE-ART PROMPT TRANSFER APPROACHES

We compare against two baselines, namely the Zero-Shot transfer by Wu et al. (2023b) and DP-OPT by Hong et al. (2023). **Zero-Shot transfer** operates in the same setup as we do and also relies on soft prompts. They perform prompt transfer by using the embeddings of some tokens from the vocabulary as a form of support vector to transform the source prompts into a relative space, and then search for the corresponding target prompt embeddings for the target model. To provide the optimal source model for their approach, we use a compressed model that we obtained by keeping the embedding layer frozen during KD (see row 3 in Table 14). **DP-OPT**, in contrast to ours, is designed for discrete prompts. They first tune a discrete prompt locally and then directly use it on the large model. Since their method relies on the small model having good performance, we execute their method in 2 setups for a fair comparison. 1) We tune their source prompt using our compressed model as the small model, and 2) we use GPT2 as the small model. The latter is expected to have significantly higher performance and yield much better prompts. To avoid the massive hyperparameter tuning required for the private tuning in DP-OPT, we resolve to the standard OPT without DP guarantees following their implementation (Hong et al., 2023). The obtained results represented an upper bound of DP-OPT, as introducing DP usually degrades performance. Our results in Table 4 highlight that our POST significantly outperforms all baselines even in the DP regime. Additional results with GPT2-XL can be found in Table 11 in the Appendix.

## 6    CONCLUSIONS

We present POST, a framework for the private transfer of soft prompts that enables adapting LLMs of an LLM provider with private user data while protecting both the user's privacy and the LLM provider's intellectual property. POST relies on distillation to enable an LLM provider to share a small model with limited utility to a client for local prompt tuning on their private data, optionally with DP guarantees. Using our new prompt transfer method that leverages a small set of public data, the LLM provider can then transfer the prompt to their model. Our experiments highlight that POST achieves significant improvements on the private tasks through the prompt transfer, improves computational efficiency of prompt tuning and outperforms all private prompt transfer baselines. Thereby, our work paves the way for a wider and more trustworthy application of LLMs.

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

## A    LIMITATIONS

Our work proposes a method to protect the privacy and confidentiality of private data during the prompt tuning phase, however, we didn't address the privacy leakage risk during the inference phase. Also, compression of the LLMs through knowledge distillation techniques may be computationally expensive for LLM providers. Additionally, in our method, the selection of a public dataset will affect the transfer performance of soft prompts. While we observe, in general, that public datasets that have a similar structure to the private data work best for transfer, there is no ideal strategy for selecting the optimal public dataset

## B    BROADER IMPACTS

Regarding the broader impacts of our work, we propose a private transfer of soft prompts from a small language model to a large LLM. The primary positive societal impact of our work is that our method can protect local data privacy and also the intelligent property of the large model provider, which encourages wider and more trustworthy applications of LLMs. Additionally, since our transfer enables more compute efficient prompt tuning and enables to re-use existing prompts, it can have a positive environmental impact.

## C    EXPERIMENTAL SETUP

### C.1    KNOWLEDGE DISTILLATION

We follow the procedure of (Sanh et al., 2019) to initialize and distill our compressed model. We use the first and last layers of Roberta-base, the first two and last two layers of GPT2-XL and the first and last layers of Llama2-7b to initialize our compressed Roberta-base, GPT2-XL and Llama2-7b before knowledge distillation. We also initialize the small student model's word embedding and language

modeling head the same as their teacher model. We conduct experiments on whether to freeze the language modeling head and/or word embedding during knowledge distillation in Appendix E.1. The model's structure and size are listed in Table 5.

Table 5: **Model size before and after distillation.**

| model | layer number | hidden dimension | head number | parameter num (M) |
|---|---|---|---|---|
| Roberta-base | 12 | 768 | 12 | 125 |
| Our distilled Roberta-base | 2 | 768 | 12 | 53 |
| GPT2-XL | 48 | 1600 | 25 | 1560 |
| Our distilled GPT2-XL | 4 | 1600 | 25 | 205 |
| Llama2-7b | 32 | 4096 | 32 | 6738 |
| Our distilled Llama2-b | 2 | 4096 | 32 | 667 |

During knowledge distillation, we use the BookCorpus (Zhu et al., 2015) dataset, and we took the checkpoint model that distilled for 50,0000 steps. The hyperparameters used in knowledge distillation are shown in Table 6.

Table 6: **Hyperparameters in knowledge distillation.**

| $\alpha_{ce}$ | $\alpha_{lm}$ | $\alpha_{cos}$ | lr | batch size |
|---|---|---|---|---|
| 5.0 | 2.0 | 1.0 | 0.00025 | 5 |

## C.2 TEXT-INFILLING TASKS

We use the text-infilling setting for the classification task. The setting is to let the model predict the ground truth text instead of using a classification head to output the class probability. To increase the robustness of this method, we use multiple ground truth text labels, and compare the average probability of outputting those text labels. See Table 7 for task templates and the ground truth labels used in our experiment.

Table 7: **Task template and ground truth labels used in text-infilling.** ¡s¿ means the sentence used in the dataset.

| Dataset | Task Template Roberta | Task Template GPT2 | Ground Truth Text Label |
|---|---|---|---|
| sst2 | ¡s¿, it was ¡mask¿ | ¡s¿, it was | 0: [" terrible"," negative"," bad"," poor"," awful"] |
| | | | 1: [" positive"," good"," great"," awesome"," brilliant"," amazing"] |
| imdb | ¡s¿, it was ¡mask¿ | ¡s¿, it was | 0: [" terrible"," negative"," bad"," poor"," awful"] |
| | | | 1: [" positive"," good"," great"," awesome"," brilliant"," amazing"] |
| tweet | ¡s¿, it was ¡mask¿ | ¡s¿, it was | 0: [" terrible"," negative"," bad"," poor"," awful"] |
| | | | 1: [" moderate"," neutral"," balanced"]] |
| | | | 2: [" positive"," good"," great"," awesome"," brilliant"," amazing"] |
| arisetv | ¡s¿, it was about ¡mask¿ | ¡s¿, it was about | 0: [" business"], 1: [" sports"], 2: [" politics"] |
| | | | 3: [" health"],4: [" entertainment"],5: [" technology"," science"] |

## C.3 PROMPT TUNING

Following (Su et al., 2022)'s setting, we use the soft prompt with a length of 100 tokens in all our experiments. We follow (Duan et al., 2024)'s setting to obtain DP private prompt with PromptDPSGD. Table 8 shows the hyperparameters used in this experiment.

## C.4 PUBLIC DATASETS FOR PROMPT TRANSFER

We rely on small public datasets to perform our prompt transfer. A question is the right choice of the public dataset. We normally choose the public dataset that performs a similar task as the private dataset, such as choosing imdb or tweet as the public dataset of sst2 as they are all sentiment classification tasks. Transferring with a public dataset that performs a different task from the private dataset may lead to suboptimal performance, we tested this setting to transfer soft prompt trained on arisetv, a topic prediction dataset. The transfer performance of using tweet as public dataset is

Table 8: **Hyperparameters used during promptDPSGD.**

| dataset | $\delta$ | epochs | lr |
|---------|----------|--------|-----|
| sst2 | $1.5 \times 10^{-5}$ | 20 | 0.1 |
| imdb | $4 \times 10^{-5}$ | 20 | 0.1 |
| tweet | $2 \times 10^{-5}$ | 20 | 0.1 |
| arisetv | $2 \times 10^{-4}$ | 40 | 0.1 |

acceptable but generally worse than using agnews, another topic prediction dataset, as a public dataset. In general, we found that the public and private dataset do not need to have the same structure, such as class number. For example, using tweet (3 classes) as a public dataset leads to better transfer performance than imdb (2 classes) on sst2 (also 2 classes). This highlights the robustness of our method and the broad selection of public datasets for the transfer.

We report the hyperparameters used in the transfer experiments as Tables 9 and 10.

Table 9: **Hyperparameters used during prompt transfer.**

| model | batch size | optimizer | lr |
|-------|-----------|-----------|-----|
| Roberta-base | 32 | Adam | 0.001 |
| GPT-XL | 8 | Adam | 0.001 |
| Llama2-7b | 4 | Adam | 0.0005 |

Table 10: **Setting of $\alpha$ for different datasets and models during prompt trasnfer.**

| model | dataset | | | |
|-------|------|------|-------|---------|
| | sst2 | imdb | tweet | arisetv |
| Roberta-base | 0.8 | 0.8 | 0.5 | 0.5 |
| GPT2-XL | 0.7 | 0.7 | 0.2 | 0.6 |
| Llama2-7b | 0.6 | 0.8 | 0.6 | 0.6 |

# D ADDITIONAL EXPERIMENTS

## D.1 BASELINE COMPARISON

We also run the baseline comparison on the GPT-XL model, we report the results in Table 11. Our method consistently outperforms other methods with this model.

## D.2 ADDTIONAL DATASETS

We present more results for additional datasets. Table 12 shows results for another classification dataset, namely MPQA, and highlights that our method outperforms the baselines significantly.

To further demonstrate the capability of our method beyond classification tasks, we conducted experiments on open-ended tasks. We evaluated our method's effectiveness on the MIT-D movie dataset consisting of 1561 train and 415 test samples. The task is to extract a movie's director from a given movie description. Instead of generating a single token in the classification tasks, this task requires generating multiple tokens with varying lengths. The result is shown below. Our method's performance (Transfer Acc) is higher than Full ZS and Compressed PT, highlighting our method's applicability to open-ended tasks.

## D.3 DISTILLATION TIME

We extended our Table 3 which conducted with GPT2-XL with the knowledge distillation time and the total runtime of our method, including KD, for arisetv and sst2 datasets in Table 13. This is the worst-case scenario where the distilled model is only used once. These results show that for standard-large datasets, our method is already faster in comparison to tuning **one single prompt on**

Table 11: **Baseline comparison.** We present the performance of our method against state-of-the-art baselines on GPT2-XL.

| Method | $\Phi_t$ | $\Phi_s$ | sst2 | imdb | tweet | arisetv |
|---|---|---|---|---|---|---|
| OPT (Hong et al., 2023) | GPT2-XL | our compressed | 60.67 | 61.70 | 30.70 | 42.87 |
| OPT (Hong et al., 2023) | GPT2-XL | GPT2 | 62.16 | 63.18 | 35.20 | 46.38 |
| Zero-Shot Transfer (Wu et al., 2023b) | GPT2-XL | our compressed | 63.65 | 61.27 | 41.60 | 56.64 |
| Zero-Shot Transfer (Wu et al., 2023b) with DP | GPT2-XL | our compressed | 63.42 | 61.71 | 41.35 | 57.25 |
| **POST (ours)** | GPT2-XL | our compressed | **85.89** | **83.93** | **61.75** | **87.56** |
| **DP-POST (ours)** | GPT2-XL | our compressed | 84.06 | 78.03 | 58.05 | 82.12 |

| Private | Full ZS | Full PT | Compressed PT | Direct Transfer | Public | Transfer Acc |
|---|---|---|---|---|---|---|
| MPQA | 46.89 | 92.36 | 83.82 | 32.96 | sst2 | **87.37** |
| MIT-D | 70.84 | 92.28 | 21.69 | 43.61 | AIE | **75.66** |

Table 12: **Confidential prompt transfer performance**. We conduct additional experiment on MMPQA and MIT-D movie dataset with Llama2-7b.

**the large model**. For small datasets, the distillation time amortizes to give our method an advantage after a few soft prompts.

# E    ADDITIONAL ABLATION EXPERIMENTS

## E.1    KNOWLEDGE DISTILLATION DESIGN

**Knowledge Distillation Setup.** We also investigated the best way of performing KD to improve prompt transferability. In particular, we analyzed the impact of keeping the word embedding or(and) language modeling heads frozen during KD on the prompt transfer performance. Our results in Table 14 highlight that keeping the language modeling head fixed performs slightly better than the alternative which mainly perform on-par. These results indicate that the successful transfer of our method is robust to the KD and independent of any specific KD setting.

## E.2    INFLUENCE OF COMPRESSED MODEL SIZE

In Table 15, we also compare the transfer performance from distilled models with different compression ratios. Based on our empirical analysis, we found as the distilled model becomes larger, the transfer performance generally becomes better. However, it also requires more distillation time and more computational resources from the user to tune the soft prompt locally. We found that our choice of 2-layer (4-layer) compressed model for Roberta-base (GPT2-XL) offers a reasonable balance between model size and performance.

To study the relationship between transfer performance (evaluated by downstream task accuracy) and performance of the compressed model (evaluated by checkpoint loss), we also conducted ablations where we compress the models to different numbers of layers layers and with different distillation steps. We present the results in Figure 6a and Figure 6b. They highlight that overall better compressed models lead to better transfer accuracy.

## E.3    TRANSFER LOSS DESIGN

We further conducted an ablation study on the effectiveness of our designed transfer loss function. The results in Table 16 show that incorporating both losses leads to better performance compared to using only the first or second loss.

We also conducted detailed ablation studies on the effect of different values of $\alpha$ from Equation (3), the results are in Table 17. Our results indicate that there is a wide range of alphas that yield comparable results, showing that our method is robust to the choice of alpha.

Table 13: Runtime for knowledge distillation (KD) and the total runtime of our method, including KD.

| Method | Runtime for arisetv (min) | Runtime for sst2 (min) |
|---|---|---|
| PT on $\Phi_t$ | 184 | 2660 |
| Knowledge Distillation | 1203 | 1203 |
| Ours total (PT on $\Phi_s$ + KD + transfer) | 1405 | 1612 |

Table 14: **Analyzing the KD setup.** We perform an ablation on different designs of the KD and present their impact on the prompt transfer for the private arisetv dataset, using agnews as public data. We analze different combinations of freezing the embedding (Fix emb) and freezing the language modeling head (Fix head).

| model | Fix emb | Fix head | Acc. | model | Fix emb | Fix head | Acc. |
|---|---|---|---|---|---|---|---|
| roberta-base | ✗ | ✓ | 81.68 ±0.764 | GPT2-XL | ✗ | ✓ | 87.52 ±0.505 |
| | ✓ | ✓ | 80.79 ±0.885 | | ✓ | ✓ | 86.51 ±0.726 |
| | ✓ | ✗ | 80.84 ±0.360 | | ✓ | ✗ | 86.81 ±0.732 |
| | ✗ | ✗ | 80.11 ±0.738 | | ✗ | ✗ | 87.48 ±0.170 |

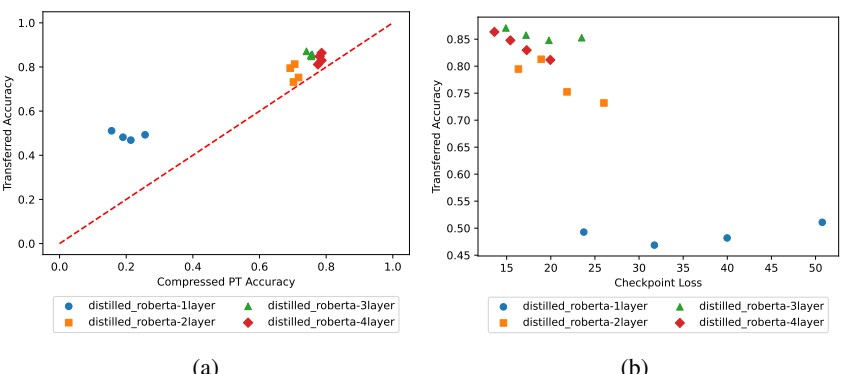

(a)                                          (b)

Figure 6: **Analysis of transferred accuracy versus compressed PT accuracy and checkpoint loss.** (a) compares transferred accuracy to compressed PT accuracy for different distilled RoBERTa models. (b) compares transferred accuracy to checkpoint loss for different distilled RoBERTa models.

Table 15: **Confidential prompt transfer performance on Roberta-base, with different numbers of layers in the compressed model.**

| # layers in distilled version | 1 layer | 2 layers (paper) | 3 layers |
|---|---|---|---|
| sst2 | 84.52 | 87.73 | 88.53 |
| imdb | 78.01 | 83.96 | 83.64 |
| tweet | 50.65 | 54.55 | 61.50 |
| arisetv | 53.62 | 82.73 | 86.45 |
| Distill time | 6h 04min | 6h 45min | 7h 35min |

Table 16: **Ablation study on the loss design, using Roberta-base with different datasets.**

| Private set | Public set for transfer | Direct Transfer | First loss term only | Second loss term only | Both loss terms |
|---|---|---|---|---|---|
| sst2 | tweet | 76.49 | 78.66 | 86.01 | 87.73 |
| imdb | tweet | 76.92 | 80.46 | 82.34 | 83.96 |
| tweet | sst2 | 43.10 | 57.05 | 49.15 | 58.25 |
| arisetv | arisetv | 47.82 | 82.00 | 60.14 | 82.73 |

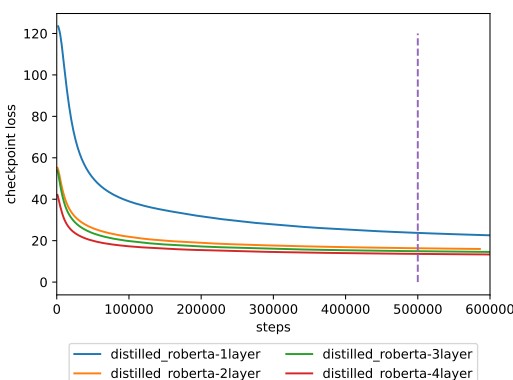

Figure 7: Checkpoint loss on training steps for distilled RoBERTa models with varying numbers of layers. The plot illustrates the convergence behavior of 1-layer, 2-layer, 3-layer, and 4-layer distilled models. The dashed vertical line represents the point at which a specific checkpoint is selected for evaluation in our experiments.

| $\alpha$ | 0.0 | 0.1 | 0.2 | 0.3 | 0.4 | 0.5 |
|---|---|---|---|---|---|---|
| SST2 | 69.57 (2.64) | 68.43 (5.40) | 70.76 (4.70) | 73.78 (2.81) | 70.18 (4.83) | 85.13 (4.34) |
| Arisetv | 81.00 (0.37) | 82.25 (0.87) | 79.83 (2.06) | 81.32 (3.56) | 82.81 (0.87) | 83.45 (0.53) |

| $\alpha$ | 0.6 | 0.7 | 0.8 | 0.9 | 1.0 |
|---|---|---|---|---|---|
| SST2 | 89.68 (0.20) | 89.18 (0.70) | 89.22 (0.53) | 88.69 (0.69) | 86.47 (1.62) |
| Arisetv | 85.47 (1.61) | 84.30 (4.75) | 83.98 (1.76) | 83.33 (2.06) | 84.02 (0.91) |

Table 17: Performance of models on SST2 and Arisetv datasets at various $\alpha$ values. Transferred accuracy are presented as mean (standard deviation).

