# OpenReview forum: "POST: A Framework for Privacy of Soft-prompt Transfer"
_ICLR.cc/2025/Conference — Submitted to ICLR 2025_

### Official Review · Reviewer_dYht · 2024-10-19

**Soundness:** 2
**Presentation:** 3
**Contribution:** 2
**Rating:** 3
**Confidence:** 4

**Summary:**

The paper studies the problem of differentially private soft prompt tranfer. They propose the POST framework which consists of 3 steps: 1) The LLM provider compresses their model into a smaller model using techniques in knowledge distillation and then sends the distilled model to the user. 2) The user performs private prompt tuning using PromptDPSGD (Duan et al., 2024) on the distilled model. 3) The user transfers this prompt on the large LLM.

**Strengths:**

The related previous work and key concepts (e.g., prompt tuning, differential privacy, knowledge distillation, prompt transfer) are clearly explained. The proposed algorithm is described in detail.

**Weaknesses:**

1, Motivation Unclear: This paper focuses on privacy-preserving soft prompt transfer, where the LLM provider is untrusted. Could you provide examples of closed-source LLM providers that support soft prompts? If, instead, the LLM provider is an open-source model that can be run locally (e.g., GPT-2 XL or Llama-2-7b, as mentioned in this paper), privacy concerns could potentially be mitigated by running the model locally. Could you clarify the motivation behind the need for privacy-preserving transfer in this context?




2, Novelty Concerns: Given the existing work for knowledge distillation (Sanh et al., 2019) and PromptDPSGD (Duan et al., 2024), what is the technical novelty in this paper?




3, Practical Concern: The proposed approach involves the LLM provider performing knowledge distillation to compress their LLM into a smaller model and then sending this distilled model to the user. Is this practical? It’s unclear if an LLM provider would be willing to share a distilled model, given proprietary concerns, and risks of potential attacks.

4, Missing DP Guarantee: I recommend including a formal theorem that states the DP guarantee of the proposed method, and a proof to ensure the privacy guarantee is rigorously supported.



5, Missing Experimental Details: The value of $\delta$ used in this paper should be mentioned in the main paper. What is the standard deviation of the results in Table 1, 2 and 4?


6, Some Typos:

(1) Line 117: fist $\rightarrow$ first.

(2) Line 119: There’s an extra comma.



(3) Line 132: from student to teacher $\rightarrow$ from teacher to student.


(4) Line 256: $\mathcal{N}(0,\sigma^2 c^2)\rightarrow \mathcal{N}(0,\sigma^2 c^2 \mathbf{I})$

**Questions:**

1. I don't understand the statement in this paper ``KD has been shown effective to compress LLMs during the pre-training phase while maintaining
their performance (Sanh et al., 2019)". The cited paper discusses BERT, which is not typically recognized as an LLM. Could you clarify this point? Additionally, can all types of LLMs (e.g., GPT, Llama, Mistral) be effectively compressed using knowledge distillation?



2. This paper considers two baselines, namely DP-OPT (Hong et al., 2023) and Zero-Shot Transfer (Wu et al., 2023b). Hong et al. (2023) use the TREC, MPQA, and Disaster datasets, and Wu et al. (2023b) consider the LAMA dataset. It is unclear why these datasets are not included in the experiments of this paper. Could the authors also include these datasets in the experiments?

---

> ### Author Response · Authors · 2024-11-24
>
> >**1, Motivation Unclear: This paper focuses on privacy-preserving soft prompt transfer, where the LLM provider is untrusted. Could you provide examples of closed-source LLM providers that support soft prompts? If, instead, the LLM provider is an open-source model that can be run locally (e.g., GPT-2 XL or Llama-2-7b, as mentioned in this paper), privacy concerns could potentially be mitigated by running the model locally. Could you clarify the motivation behind the need for privacy-preserving transfer in this context?**
>
> Currently, many closed-source LLM providers do not support soft prompts, primarily because the computational overhead associated with implementing and managing such functionality is prohibitively expensive. Our method addresses this gap by significantly reducing the computational overhead required for soft prompt tuning by first tuning the soft prompt on a small model and then transferring it to the large model.
>
> As shown in Table 3 in the paper, *our approach reduces computational overhead and thus has the potential to enable this new functionality* in closed-source LLMs, unlocking a broader range of applications. Running large open-source models like Llama-2-7b locally can indeed mitigate certain privacy concerns, as sensitive data remains within the user's infrastructure. However, deploying and maintaining these models locally presents high computational and storage requirements, as well as the additional engineering and operational efforts, that may be prohibitive for some users. Our work addresses this problem.
>
> >**2, Novelty Concerns: Given the existing work for knowledge distillation (Sanh et al., 2019) and PromptDPSGD (Duan et al., 2024), what is the technical novelty in this paper?**
>
> We would like to emphasize that *no work, so far, has successfully transferred soft-prompts* while maintaining the privacy of user data. Our work is the first to offer a practical solution to this important problem by thoughtfully integrating several key components including knowledge distillation, soft-prompt fine-tuning and transfer. Additionally, our method achieves supervisor utility and privacy protection compared to state-of-the-art prompt transfer methods [Hong et al., 2023; Wu et al., 2023].
>
> >**3, Practical Concern: The proposed approach involves the LLM provider performing knowledge distillation to compress their LLM into a smaller model and then sending this distilled model to the user. Is this practical? It’s unclear if an LLM provider would be willing to share a distilled model, given proprietary concerns, and risks of potential attacks**
>
> The distilled models resulting from our KD process are significantly smaller (plesae see Table 5 in the paper) and possess limited capabilities compared to the original LLMs (please see compressed PT in tables 1 and 2). This reduction in size and functionality ensures that the core intellectual property and advanced functionalities of the original models remain protected. By sharing a model with reduced utility, providers can mitigate the risk of exposing proprietary information inherent in the larger models.
>
> >**4, Missing DP Guarantee: I recommend including a formal theorem that states the DP guarantee of the proposed method, and a proof to ensure the privacy guarantee is rigorously supported.**
>
> We apply PromptDPSGD (Duan, 2023) to tune the soft prompt with DP guarantees on the small LLM. Analogous to the work by Duan et al., our privacy guarantees follow immediately from the privacy analysis of DPSGD.
>
> >**5, Missing Experimental Details: The value of δ used in this paper should be mentioned in the main paper. What is the standard deviation of the results in Table 1, 2 and 4?**
>
> We use δ as 1 over the size of each dataset, this is commonly used in most of the DP related works (including Duan et al. 2023). We ran the experiment 3 times and reported the best transferred accuracy in the tables.
>
> >**6. Some Typos**
>
> Thank you for taking the time to review our paper carefully. We have addressed the typos on our end in the revised version. We would also like to clarify that the statement in line 132 is correct, as our intention is to transfer the *prompt* from the student to the teacher.

---

> ### Author Response · Authors · 2024-11-24
>
> >**I don't understand the statement in this paper ``KD has been shown effective to compress LLMs during the pre-training phase while maintaining their performance (Sanh et al., 2019)". The cited paper discusses BERT, which is not typically recognized as an LLM. Could you clarify this point? Additionally, can all types of LLMs (e.g., GPT, Llama, Mistral) be effectively compressed using knowledge distillation?**
>
> Sanh et al. (2019) discussed distillation for BERT, indeed. While not typically classified as a large language model (LLM) in the context of models like GPT, LLaMA, or Mistral, BERT is a pre-trained transformer which is a backbone of modern LLMs. Our intention was to highlight the general effectiveness of knowledge distillation (KD) in compressing this types of models.  Recent research has extended KD techniques to compress larger models. For instance, MiniLLM (Gu et al., 2024)[1] explores effective KD approaches for large LLMs including LLAMA-7B, LLAMA-13B, and OPT-13B. (Sreenivas et al., 2024)[2] compress the Llama 3.1 8B and Mistral NeMo 12B with pruning and knowledge distillation. Thus, the statement holds for LLMs in general. We added the other citations in the revised version of the paper to support our claim.
>
> >**This paper considers two baselines, namely DP-OPT (Hong et al., 2023) and Zero-Shot Transfer (Wu et al., 2023b). Hong et al. (2023) use the TREC, MPQA, and Disaster datasets, and Wu et al. (2023b) consider the LAMA dataset. It is unclear why these datasets are not included in the experiments of this paper. Could the authors also include these datasets in the experiments?**
>
> We did not use the LAMA dataset because it is a knowledge-probing dataset designed to evaluate whether a model contains commonsense knowledge. Since commonsense knowledge is generally publicly known and does not qualify as private information requiring protection, we exclude the LAMA dataset from our analysis.
>
> As requested, we add the results for MPQA. We are currently running the other datasets and, in case of acceptance, will include them in the camera-ready version. On MPQA, our method is successful and provides a significantly higher transfer accuracy than Compressed PT Acc, and then the zero shot base line.
>
> Beyond the classification tasks, we also ran experiments on open-ended Q&A tasks which are not shown in either DP-OPT (Hong et al., 2023) or in Zero-Shot Transfer (Wu et al., 2023b). Instead of generating a single token as in the classification tasks, this task requires generating multiple tokens with varying lengths. The result is also shown in the table below.  Our method’s performance (Transfer Acc) is higher than Full ZS and Compressed PT, highlighting our method’s applicability to open-ended tasks.
>
> We also add those results as Table 12 in the appendix of the revised version of the paper.
>
> | Private  | Full ZS | Full PT | Compressed PT | Direct Transfer | Public | Transfer Acc |
> |----------|---------|---------|---------------|-----------------|--------|--------------|
> | MPQA    | 46.89  | 92.36   | 83.82         | 32.96           | sst2    | **87.37**        |
> | MIT-D    | 70.84   | 92.28   | 21.69         | 43.61           | AIE    | **75.66**       |
>
> **References:**
>
> [1] *”MiniLLM: Knowledge Distillation of Large Language Models.”* Yuxian Gu, Li Dong, Furu Wei, Minlie Huang. International Conference on Learning Representations (ICLR), 2024.
>
> [2] *”LLM Pruning and Distillation in Practice: The Minitron Approach.”* Sharath Turuvekere Sreenivas, Saurav Muralidharan, Raviraj Joshi, Marcin Chochowski, Mostofa Patwary, Mohammad Shoeybi, Bryan Catanzaro, Jan Kautz, Pavlo Molchanov. arXiv:2408.11796 2024.

---

> > ### Comment · Reviewer_dYht · 2024-11-24
> >
> > Thank you for the detailed response and the effort put into additional experiments. The responses have addressed some of my concerns. However, the motivation for the problem studied in the paper remains unclear to me. It is still not evident why developing privacy-preserving transferable soft prompts for closed-source LLMs that currently do not support soft prompts is important. Additionally, the technical novelty appears limited, even though the paper is the first to study this problem. Therefore, I keep my original score unchanged.

---

> ### Author Response · Authors · 2024-11-24
> **Thank you for your response**
>
> Thank you for taking the time to review our work. We appreciate your acknowledgment that some of your concerns have been addressed.
>
> >**However, the motivation for the problem studied in the paper remains unclear to me. It is still not evident why developing privacy-preserving transferable soft prompts for closed-source LLMs that currently do not support soft prompts is important.**
>
> Our work directly addresses the limitations of closed-source LLMs, which currently lack support for soft prompts. We also provide an in-depth analysis of how our method can be adopted by closed LLM providers, as detailed in the results below.
>
> Additionally, the motivation behind our work and novelty are clearly recognized by the other reviewers. Reviewer y96m remarked, "**The setting of the paper is well-motivated**." Similarly, Reviewer mvG3 acknowledged, "The paper brought focus on the privacy and efficiency issue on prompt tuning." Importantly, our method also addresses the broader challenge of reducing the cost of transferring prompts between LLMs. Reviewer QrLz highlighted the significance of our contribution, stating, "The paper studies a very interesting problem and **proposed a novel framework** to allow users to 'perform soft prompt tuning' without revealing their private data and LLM providers not revealing their protected LLMs. **The most novel part comes from privacy-preserving prompt transfer through public data, which largely protects data privacy while maintaining inference performance.**"
>
> We hope these perspectives from other reviewers and this additional context help clarify the value and significance of our work.
>
> >**Additionally, the technical novelty appears limited, even though the paper is the first to study this problem.**
>
> **Our method consistently outperforms all prior approaches to prompt transfer across a variety of tasks and models!** To further validate our findings, we conducted additional experiments using Llama2-7b. The results, presented in the table below (and included in the main paper as Table 4), confirm that our method surpasses all baselines by a significant margin! This indicates the technical novelty of our method.
>
> | **Method** | **Φₜ** | **Φₛ** | **SST2** | **IMDB** | **Tweet** | **arisetv** |
> |------------------------------------------------|------------|-----------------|--------|--------|--------|---------|
> | OPT [Hong et al., 2023] | Llama2-7b | our compressed | 81.31| 67.40 | 26.90 | 82.00 |
> | OPT [Hong et al., 2023] | Llama2-7b | GPT-2 | 81.65| 62.93 | 41.15 | 78.26 |
> | Zero-Shot Transfer [Wu et al., 2023b] | Llama2-7b | our compressed | 62.38 | 70.57  | 42.80 | 58.33 |
> | Zero-Shot Transfer [Wu et al., 2023b] with DP | Llama2-7b | our compressed | 53.55 | 69.47 | 41.65 | 59.54 |
> | **POST (ours)** | Llama2-7b | our compressed | **90.14** | **86.27** | **61.70** | **86.71** |
> | **DP-POST (ours)** | Llama2-7b | our compressed | 89.91 | 83.26 | 59.55 | 82.60 |
>
> Moreover, our method is very efficient. We extended Table 3 (GPT2-XL) to include compute times, with the new results detailed in the table below and Table 13 of the appendix. For the relatively larger SST2 dataset, prompt-tuning on the proxy model takes 310 minutes, and transferring the tuned soft prompt to the large LLM takes an additional 99 minutes, resulting in a total of 409 minutes. In contrast, directly fine-tuning a soft prompt on the large LLM requires 2660 minutes. A similar efficiency trend is observed across other datasets. For the smaller arisetv dataset, the process is faster. Prompt-tuning on the smaller model takes just 23 minutes, and the transfer step adds 99 minutes, totaling 122 minutes—over an hour faster than the 184 minutes required for direct prompt-tuning on the large teacher LLM. These results clearly demonstrate the substantial time savings our method provides.
>
> | **Method** | **Runtime for arisetv (min)** | **Runtime for sst2 (min)** |
> |----------------------|---------------------------|-------------------------|
> | PT on Φ_t | 184 | 2660 |
> | (1) PT on Φ_s | 23 | 310 |
> | (2) Transfer | 99 | 99 |
> | **Ours total (1)+(2)** | **122** | **409** |
>
> >**Therefore, I keep my original score unchanged.**
>
> We thank the reviewer for their thoughtful feedback, which has greatly contributed to improving the quality of our paper. We hoped that the new results, demonstrating superior performance compared to all baselines and achieving this with minimal computational overhead, might encourage a reconsideration and potentially a higher score.

---

> > ### Author Response · Authors · 2024-12-01
> > **Have concerns been addressed?**
> >
> > We would like to follow up on our responses, especially on the motivation for the paper and technical novelty.
> >
> > 1. The current methods enable the transfer and privacy of discrete prompts for closed LLMs, but soft prompts significantly outperform discrete prompts when used with open LLMs [1]. Therefore, our objective is to extend these capabilities to support the transfer and privacy of soft prompts for closed LLMs, bridging the gap in privacy, performance, and computational efficiency.
> > 2. Our method consistently outperforms all prior approaches to prompt transfer across a variety of tasks and models. The *"most novel part comes from privacy-preserving prompt transfer through public data"* (Reviewer QrLz). We designed a novel framework for soft prompt transfer. We selectively combine strong elements (aggressive knowledge distillation using fixed language modeling head and PromptDPGSG) with privacy-preserving soft prompt transfer that aligns predictions and directions of change between the distilled student model and the final target model.
> >
> > We thank the Reviewer for their thoughtful feedback, which has greatly contributed to improving the quality of our paper. We hope that our responses encourage a reconsideration and a higher score for this work.
> >
> > **References:**
> >
> > [1] *"Open LLMs are Necessary for Current Private Adaptations and Outperform their Closed Alternatives"* Vincent Hanke, Tom Blanchard, Franziska Boenisch, Iyiola Emmanuel Olatunji, Michael Backes, Adam Dziedzic. NeurIPS 2024. https://openreview.net/forum?id=Jf40H5pRW0

---

### Official Review · Reviewer_QrLz · 2024-11-01

**Soundness:** 3
**Presentation:** 3
**Contribution:** 3
**Rating:** 6
**Confidence:** 3

**Summary:**

In this paper, the authors proposed a new framework that allows the users to perform private soft prompt tuning without accessing the original protected LLMs without leaking their private data. They proposed to  first obtained a distilled model from the original LLM, a step needed from the LLM provider, and then the users will utilize this distilled model to perform soft prompt tuning and transfer the resulting soft prompt to prompt the original LLM. They conducted a few experiments to demonstrate the performance of their proposed framework and compare it with several baselines.

**Strengths:**

The paper studies a very interesting problem and proposed a novel framework to allow users to “perform soft prompt tuning” without revealing their private data and LLM providers not revealing their protected LLMs. The most novel part comes from privacy-preserving prompt transfer through public data, which largely protect data privacy and maintain inference performance.

**Weaknesses:**

I listed a few weaknesses below:
1. It is unclear how to select public dataset for prompt transfer, there is no guidance on how to select such a public dataset and how would the selection of public dataset affect the performance. From their experimental results in Table 1, it seems like different selection would make a large enough impact on the inference accuracy.
2. No explanation on what would be the termination criterion for knowledge distillation step. The authors described that the distilled model should behave closely to the original model but not meeting the actual inference performance of the original model, which is too vague. I do not see either the ablation studies on how to select $\alpha_{ce}$, $\alpha_{lm}$ and $\alpha_{cos}$.

Overall, the main concern is that the proposed method contains many steps involving hyperparameters and public datasets need to be tuned and selected. More ablation studies and further theoretical analysis are needed to understand the general practical performance of the proposed method.

**Questions:**

1.	No ablation study is found for the selection of different value of $\alpha$ in Eq. (3).
2.	How to select a public dataset used for prompt transfer? From Table 1, it seems like sometimes selecting a different public dataset affects the final test accuracy a lot.
3.	Other than next word prediction task, does the proposed method perform well on other language task? For example, question answering and reading comprehension?
4.	Why there is no result for llama2-7b in Table 2?
5.	When comparing to the status of the art baselines, it seems like larger models have better zero-shot performance. In this case, in Table 4, can you include more results on larger models like llama2?
6.	From Fig. 5, it seems like the attack success rate is low even without DP. In this case, would that be the MIA algorithm used in the paper is not good enough? It seems like there is no motivation to add DP to protect private data.
7.	If the original LLM is mixture of expert model, does the proposed framework apply to that case?

---

> ### Author Response · Authors · 2024-11-24
>
> >**It is unclear how to select public dataset for prompt transfer, there is no guidance on how to select such a public dataset and how would the selection of public dataset affect the performance. From their experimental results in Table 1, it seems like different selection would make a large enough impact on the inference accuracy. How to select a public dataset used for prompt transfer? From Table 1, it seems like sometimes selecting a different public dataset affects the final test accuracy a lot.**
>
> Our results indicate that public datasets from the same *task family*, e.g., news classification, act as suitable public datasets. Notably, our experiments also highlight that the public task can have a different number of classes than the private one. We observe a dependence on the distribution shift between the private data and the public data used for the transfer, where the closer the distributions, the higher the accuracy gain.
>
> >**No explanation on what would be the termination criterion for knowledge distillation step. The authors described that the distilled model should behave closely to the original model but not meeting the actual inference performance of the original model, which is too vague.**
>
> As the termination criterion for the distillation step, we select the checkpoint directly after the loss plateaus. Through this simple approach, we obtain good performance while minimizing the distillation overhead. Please see Figure 7 in the Appendix (in the revised submission) that shows the relationship between distillation loss and steps.
>
> We conduct additional experiments on the tradeoff between distilled model and transfer performance. We present the results in Figure 6 in the updated version of the paper and also include a table here below for the Reviewer’s convenience. We distilled 4 different sizes of roberta-base and used the checkpoints at 50k, 100k, 200k, and 500k step to perform the transfer. The results highlight that generally the better the proxy model, the better the transferred performance. In the table below, we report Transferred Acc (Proxy model Acc). Each value represents the transferred accuracy (the Transfer Acc. represented by all values not in parentheses) and the accuracy on the proxy model (Proxy model Acc, for all values in the parentheses).
>
> | checkpoint_model         | 50000         | 100000        | 200000        | 500000        |
> |:-------------------------|:--------------|:--------------|:--------------|:--------------|
> | distilled_roberta-1layer | 51.09 (15.61) | 48.19 (19.02) | 46.86 (21.39) | 49.28 (25.66) |
> | distilled_roberta-2layer | 73.19 (70.18) | 75.24 (71.71) | 81.28 (70.56) | 79.47 (69.21) |
> | distilled_roberta-3layer | 85.27 (75.41) | 84.78 (75.59) | 85.75 (75.77) | 87.08 (74.09) |
> | distilled_roberta-4layer | 81.16 (77.54) | 82.97 (78.61) | 84.78 (78.05) | 86.35 (78.63) |
>
> The results show that, in general, the higher quality of the proxy model results in the higher the transferred performance (and the trend is strengthened with more layers in the proxy model).
>
> >**I do not see either the ablation studies on how to select $\alpha_{ce}$, $\alpha_{lm}$, and $\alpha_{cos}$.**
>
> We used the fixed values for the parameters $\alpha_{ce}$, $\alpha_{lm}$, and $\alpha_{cos}$ according the the DistillBERT paper [1], i.e., we used the exact parameters as indicated by the first author of the paper here: https://discuss.huggingface.co/t/reproducing-distilroberta/5217
>
> >**No ablation study is found for the selection of different value of $\alpha$ in Eq. (3).**
>
> We added an ablation study of different values of $\alpha$, which we repeated 3 times each, using Llama2.  We show the results below and also in Table 17 in the Appendix of the revised paper:
>
> |Dataset | 0.0 |  0.1 | 0.2 | 0.3 | 0.4 | 0.5 | 0.6 | 0.7 | 0.8 | 0.9 | 1.0 |
> |--------|--------|---------|--------|----------|--------|-----------|-------------|-------------|-------------|-------------|-------------|
> | SST2 | 69.57 (2.64) | 68.43 (5.40) | 70.76 (4.70) | 73.78 (2.81) | 70.18 (4.83) | 85.13 (4.34) | 89.68 (0.20) | 89.18 (0.70) | 89.22 (0.53) | 88.69 (0.69) | 86.47 (1.62) |
> | Arisetv | 81.00 (0.37) | 82.25 (0.87) | 79.83 (2.06) | 81.32 (3.56) | 82.81 (0.87) | 83.45 (0.53) | 85.47 (1.61) | 84.30 (4.75) | 83.98 (1.76) | 83.33 (2.06) | 84.02 (0.91) |
>
> Our results indicate that there is a wide range of values for the hyper-parameter $\alpha$ that yield comparable results, showing that our method is robust to the choice of $\alpha$. Additionally, we observe that combining both losses yields higher performance than using just a single loss (values of $\alpha$-s equal 0 or equal to 1).

---

> ### Author Response · Authors · 2024-11-24
>
> >**Other than next word prediction task, does the proposed method perform well on other language task? For example, question answering and reading comprehension?**
>
> We thank the Reviewer for the suggestion. Given that we model our next word prediction as a text-infiling task, this makes our approach inherently suitable to address other tasks as well. Thus, **we performed new experiments on open-ended Q&A tasks**. We evaluated our method’s effectiveness on the MIT-D movie dataset consisting of 1561 train and 415 test samples. The task is to extract a movie’s director from a given movie description. Instead of generating a single token in the classification tasks, this task requires generating multiple tokens with varying lengths. The result is shown below as well as in Table 12 in the appendix. Our method’s performance (Transfer Acc) is higher than Full ZS (Zero-Shot) and Compressed PT (Prompt Tuning), highlighting our method’s applicability to open-ended tasks.
>
> | Private  | Full ZS | Full PT | Compressed PT | Direct Transfer | Public | Transfer Acc |
> |----------|---------|---------|---------------|-----------------|--------|--------------|
> | MIT-D    | 70.84   | 92.28   | 21.69         | 43.61           | AIE    | 75.66       |
>
>
> >**Why there is no result for llama2-7b in Table 2?**
>
> We added the table as Table 2(c)  in the revised version of the paper. We also present the table here.
>
> | Private  | Full ZS | Full PT | Compressed PT | Direct Transfer | Public | Transfer Acc | Public | Transfer Acc |
> |----------|---------|---------|---------------|-----------------|----------|----------|----------|----------|
> | sst2     | 78.67   | 90.60   | 70.99         | 53.55           | tweet    | 87.50    | imdb     | 89.91    |
> | imdb     | 83.74   | 91.47   | 70.26         | 68.61           | tweet    | 82.14    | sst2     | 83.26    |
> | tweet    | 44.50   | 62.40   | 48.16         | 41.65           | imdb     | 56.60    | sst2     | 59.55    |
> | arisetv  | 76.57   | 83.73   | 64.43         | 64.73           | agnews   | 82.60    | tweet    | 75.24    |
>
>
> >**When comparing to the status of the art baselines, it seems like larger models have better zero-shot performance. In this case, in Table 4, can you include more results on larger models like llama2?**
>
> We have conducted additional experiments to address this comment, The results are listed below. We include these in the main paper now as Table 4 while moving the GPT2 results to the appendix as Table 11.  The results demonstrate that with the larger Llama2-7b, our method also outperforms baseline methods.
>
> | **Method** | **Φₜ** | **Φₛ** | **SST2** | **IMDB** | **Tweet** | **arisetv** |
> |------------------------------------------------|------------|-----------------|--------|--------|--------|---------|
> | OPT [Hong et al., 2023] | Llama2-7b | our compressed | 81.31| 67.40 | 26.90 | 82.00 |
> | OPT [Hong et al., 2023] | Llama2-7b | GPT-2 | 81.65| 62.93 | 41.15 | 78.26 |
> | Zero-Shot Transfer [Wu et al., 2023b] | Llama2-7b | our compressed | 62.38 | 70.57  | 42.80 | 58.33 |
> | Zero-Shot Transfer [Wu et al., 2023b] with DP | Llama2-7b | our compressed | 53.55 | 69.47 | 41.65 | 59.54 |
> | **POST (ours)** | Llama2-7b | our compressed | **90.14** | **86.27** | **61.70** | **86.71** |
> | **DP-POST (ours)** | Llama2-7b | our compressed | 89.91 | 83.26 | 59.55 | 82.60 |
>
>
> >**From Fig. 5, it seems like the attack success rate is low even without DP. In this case, would that be the MIA algorithm used in the paper is not good enough? It seems like there is no motivation to add DP to protect private data.**
>
> We would like to point out that we employed a simple threshold-based MIA for our experiments. There are significantly stronger MIAs, such as the Likelihood Ratio Attack (LiRA) [2] and Robust Membership Inference Attack (RMIA) [3], which would report higher risks. However, every attack-based evaluation is only as strong as the attack and might *severely* underreport the actual risk. Therefore, it is crucial to apply DP, which gives **provable privacy guarantees** against any attacker with background knowledge. Therefore, the integration of DP into our framework serves a critical role.

---

> ### Author Response · Authors · 2024-11-24
>
> >**If the original LLM is mixture of expert model, does the proposed framework apply to that case?**
>
> Our current experiments do not encompass MoE models, aligning with the settings of our baseline comparisons. However, our framework is architecture agnostic and applicable to MoE models as well. Recent research has demonstrated methods for distilling MoE models into more compact forms, for instance, the effective techniques proposed by Salians et al. [4]. We recognize the importance of this extension and plan to explore the integration of MoE models into our framework.
>
> **References:**
>
> [1] “DistilBERT, a distilled version of BERT: smaller, faster, cheaper and lighter”
> Victor Sanh, Lysandre Debut, Julien Chaumond, Thomas Wolf.
>
> [2] *”Membership Inference Attacks From First Principles.”* Nicholas Carlini, Steve Chien, Milad Nasr, Shuang Song, Andreas Terzis, Florian Tramer. IEEE Symposium on Security and Privacy (S&P) 2022.
>
> [3] *“Low-Cost High-Power Membership Inference Attacks.”* Sajjad Zarifzadeh, Philippe Liu, Reza Shokri. International Conference on Machine Learning (ICML), 2024.
>
> [4] Felipe Cruz Salinas, Kenichi Kumatani, Robert Gmyr, Linquan Liu, Yu Shi. Knowledge Distillation for Mixture of Experts Models in Speech Recognition. 2022

---

> ### Comment · Reviewer_QrLz · 2024-11-25
> **Thanks for the responses!**
>
> Thank you for the response. I think it cleared out most of my concerns related to experimentation. In terms of the ablation study on $\alpha$ in Eq. (3), it looks like at least for SST2 dataset, the accuracy changes a lot depending on what value one selects on $\alpha$. So I am a bit skeptical about the sensitivity of hyperparameter selection and how would the algorithm generalized to other real-world datasets and tasks. So I will keep my current score.

---

> > ### Author Response · Authors · 2024-11-26
> > **Thank you, the selection of $\alpha$, and other real-world datasets**
> >
> > We thank the reviewer for their positive feedback and for engaging with us in the discussion! We are also glad that our previous answers addressed most of the concerns.
> >
> > >**Hyperparameter $\alpha$**
> >
> > We did our due diligence and scrutinized **all** possible values of the hyperparameter $\alpha$. It is expected that with the whole spectrum, some of these values are significantly worse than others. The sharp increase in the performance from $\alpha=0.4$ to $\alpha=0.5$ and then the best performance of the algorithm with $\alpha=0.6$ for both SST2 and Arivsetv datasets suggests that both loss terms from Equation (3) are necessary to obtain good performance and the second part of the loss is slightly more helpful.
> >
> > >**Other real-world datasets**
> >
> > We note that the IMDB dataset is very similar in terms of the task but much larger on disk than SST2. To be precise, SST2 contains around 70k pithy expert movie reviews [1] (short single-sentence reviews) while IMDb [2] contains 50k full-length lay movie reviews. The task in both cases is a binary classification of the sentiment for movie reviews. In fact, both datasets were used interchangeably with training on one dataset and evaluating on the other dataset, and vice versa [3]. IMDb is a much bigger dataset of size 66.21MB, since it has longer reviews, than the SST dataset of size only 4.5 MB. We showed that our algorithm works very well on IMDB, thus, it generalizes to other real-world datasets and tasks. Additionally, we used the datasets which were leveraged in the previous work. Notably, our method consistently outperforms all prior approaches to prompt transfer across a variety of datasets and models. Does the Reviewer have any preference on which other datasets we should evaluate our algorithm on?
> >
> > **References:**
> >
> > [1] *"Recursive deep models for semantic compositionality over a sentiment tree-bank."* Richard Socher, Alex Perelygin, Jean Wu, Jason Chuang, Christopher D Manning, Andrew Ng, and
> > Christopher Potts. EMNLP 2013.
> >
> > [2] *"Learning word vectors for sentiment analysis."* Andrew L Maas, Raymond E Daly, Peter T Pham, Dan Huang, Andrew Y Ng, and Christopher Potts. ACL 2011.
> >
> > [3] *"Pretrained Transformers Improve Out-of-Distribution Robustness"*. Dan Hendrycks, Xiaoyuan Liu, Eric Wallace, Adam Dziedzic, Rishabh Krishnan, Dawn Song. ACL 2020.

---

> > > ### Author Response · Authors · 2024-11-28
> > > **Membership inference with the Likelihood Ratio Attack (LiRA)**
> > >
> > > >**From Fig. 5, it seems like the attack success rate is low even without DP. In this case, would that be the MIA algorithm used in the paper is not good enough? It seems like there is no motivation to add DP to protect private data.**
> > >
> > > We conducted an additional experiment using the stronger Membership Inference Attack (MIA) technique, specifically the Likelihood Ratio Attack (LiRA) [1]. This evaluation was performed on the SST2 dataset using 8 reference models. The results indicate a non-trivial level of data leakage, with an AUC of 0.5704 for the non-protected adaptation. By comparison, the AUC decreases to 0.5316 when applying Differential Privacy (DP) with $\varepsilon=8$, demonstrating a reduction in leakage risk compared to the baseline AUC of 0.5.
> > >
> > > It is important to note that the strength of the attack employed limits the reliability of any attack-based evaluation, and weaker attacks can significantly underestimate the true risk. This underscores the necessity of employing DP, which provides provable privacy guarantees irrespective of an attacker's strength. Consequently, integrating DP into our framework is critical for ensuring robust privacy protection.
> > >
> > > | $\varepsilon$ | LiRA AUC |
> > > |--------|-------------|
> > > | $\infty$ | 0.5704 |
> > > | 8 | 0.5316 |
> > >
> > > **References:**
> > >
> > > [1] ”Membership Inference Attacks From First Principles.” Nicholas Carlini, Steve Chien, Milad Nasr, Shuang Song, Andreas Terzis, Florian Tramer. IEEE Symposium on Security and Privacy (S&P) 2022.

---

> ### Author Response · Authors · 2024-12-01
> **Have concerns been addressed?**
>
> We would like to follow up on our responses, especially on the experiments for the $\alpha$ parameter, other real-world datasets, and the stronger membership inference attack.
>
> 1. The analysis of the parameter $\alpha$ shows that both loss terms from Equation (3) are necessary to obtain good performance and the second part of the loss (to align the direction change induced by the private prompt) is slightly more helpful.
> 2. We demonstrated that our algorithm works very well on other real-world datasets like IMDB.
> 3. We also conducted an additional experiment using the strong Likelihood Ratio Attack (LiRA) for membership inference and showed that our method reduces the privacy leakage.
>
> Do the results adequately address the reviewer's concerns?

---

### Official Review · Reviewer_mvG3 · 2024-11-03

**Soundness:** 2
**Presentation:** 2
**Contribution:** 3
**Rating:** 5
**Confidence:** 4

**Summary:**

The authors propose a framework for Privacy Soft-Prompt transfer to reduce the computation cost and potential privacy leakage. The framework mainly contains three steps as deriving SLM using knowledge distillation, locally prompt tuning, and prompt transferring using public dataset.

**Strengths:**

1. The author proposed a novel framework for transferring soft prompts between SLM and LLM, and achieved completive results on the test dataset
2. The paper brought focus on the privacy and efficiency issue on prompt tuning, and tried to reduce the privacy leakage and computational cost.

**Weaknesses:**

1. The paper writing needs to be improved. For example, the author does not clearly represent whether POST needs the original input x during the inference. If so, this may lead to privacy concerns. I mean, in tasks like classification, the LM input contains privacy information.
2. The paper needs to be clear about the setting. For black-box LMs, they cannot provide the service as giving a SLM through KD. If the author means for white-box LMs, they author should further explain why there is privacy concerns.
3. For the experiment, I think other tasks in spite of classification help improve the credibility. For tasks as classification, the LM input contains privacy information which violates the setting in this paper. For other tasks as Q&A, the soft prompt may contains privacy information and the model inputs may have less privacy concerns, which is consistent with the setting in this paper.

**Questions:**

1. I wonder whether the framework needs the original input during evaluation. If so, the model input may contains privacy information.
2. For the setting, could you please give further explanation about whether it is white or black box?
3. Could you please provide more evidence (citations, or experiments) to confirm the claim“soft prompts are highly coupled to the LLM they were tuned on, making them difficult to transfer”?

---

> ### Author Response · Authors · 2024-11-24
>
> >**For example, the author does not clearly represent whether POST needs the original input x during the inference. If so, this may lead to privacy concerns. I mean, in tasks like classification, the LM input contains privacy information. I wonder whether the framework needs the original input during evaluation. If so, the model input may contains privacy information**
>
> Our method is proposed to protect the data during the **training** phase. The user still needs to send its input to the LLM provider for inference. Private inference is an orthogonal research direction to this paper.
>
> However, our framework can integrate differentially private (DP)  text generation (text-to-text privatization) techniques [1] to address concerns related to private inference. Specifically, in private inference, our approach would generate a privatized version of the querying party’s private data with DP guarantees for use in the inference process.
>
> >**The paper needs to be clear about the setting. For black-box LMs, they cannot provide the service as giving a SLM through KD. If the author means for white-box LMs, they author should further explain why there is privacy concerns. For the setting, could you please give further explanation about whether it is white or black box?**
>
> We are happy to clarify the misunderstanding: In our work, *the LLM provider performs the KD* (knowledge distillation). The LLM provider is at the same time also the owner of the model and, thereby, has naturally white-box access. However, due to concerns for their intellectual property, the LLM provider does not want to share the model directly (in the white-box manner) with the data owners. Therefore, the model provider distills a less powerful version of the large LLM for the data owners who use this small version (white-box) for training their prompt. However, the large LLM has a **black-box access perspective for the data owners (users)**.
>
> Through our private prompt transfer, we mitigate the risk that the model provider could potentially extract sensitive information about the data owner from the trained prompt.
>
> >**For the experiment, I think other tasks in spite of classification help improve the credibility. For tasks as classification, the LM input contains privacy information which violates the setting in this paper. For other tasks as Q&A, the soft prompt may contains privacy information and the model inputs may have less privacy concerns, which is consistent with the setting in this paper.**
>
> We thank the Reviewer for the suggestion. Although in the original submission, we presented results on classification tasks, we format the classification task as a text-infiling task which generates the next word prediction. This makes our approach inherently suitable to address other tasks as well. Thus, as requested, **we performed new experiments on open-ended Q&A tasks**. We evaluated our method’s effectiveness on the MIT-D movie dataset consisting of 1561 train and 415 test samples. The task is to extract a movie’s director from a given movie description. Instead of generating a single token in the classification tasks, this task requires generating multiple tokens with varying lengths. The result is shown in the table below as well as in Table 12 in the appendix of the revised submission. Our method’s performance (Transfer Acc) is higher than Full ZS (Zero-Shot) and Compressed PT (Prompt Tuning), highlighting our method’s applicability to open-ended tasks.
>
> | Private  | Full ZS | Full PT | Compressed PT | Direct Transfer | Public | Transfer Acc |
> |----------|---------|---------|---------------|-----------------|--------|--------------|
> | MIT-D    | 70.84   | 92.28   | 21.69         | 43.61           | AIE    | **75.66**     |

---

> ### Author Response · Authors · 2024-11-24
>
> >**Could you please provide more evidence (citations, or experiments) to confirm the claim“soft prompts are highly coupled to the LLM they were tuned on, making them difficult to transfer”?**
>
> The paper by Liu et al. (NeurIPS 2022) [2] that we cite in our work shows that gradient-based adaptations, such as PEFT methods, provides evidence that soft prompts are highly coupled to the LLM they are tuned on. This is because each model's embedding space is shaped by its unique training data, architecture, and optimization processes. Consequently, the same token may be represented differently across models, leading to variations in semantic interpretations.
> For this reason, our experiments demonstrate that directly transferring soft prompts between models results in suboptimal performance.
>
> Additionally, soft prompts *dimensionality* is tightly coupled to the LLM they were tuned on: their embeddings must align with the model's embedding dimension to be integrated effectively. However, embedding dimensions vary across models. For instance: RoBERTa-base: 768, RoBERTa-large: 1024, GPT-2 small: 768, GPT-2 large: 1024, Llama2-7b: 4096, Due to these differences, a soft prompt trained on one model cannot be directly applied to another with a different embedding size, as the dimensional mismatch would prevent proper integration.
>
> **References:**
>
> [1] *“Locally differentially private document generation using zero shot prompting.”* Saiteja Utpala, Sara Hooker, Pin-Yu Chen. Empirical Methods in Natural Language Processing (EMNLP), 2023.
>
> [2] *“Few-shot parameter-efficient fine-tuning is better and cheaper than in-context learning.”* Haokun Liu, Derek Tam, Mohammed Muqeeth, Jay Mohta, Tenghao Huang, Mohit Bansal, and Colin A Raffel.  Advances in Neural Information Processing Systems (NeurIPS), 2022.

---

> > ### Comment · Reviewer_mvG3 · 2024-11-25
> > **Thanks for your response**
> >
> > Thank you for the detailed response. However, I remain concerned that the classification task is of little value here and suggest adding more non-classification experiments. Additionally, comparing only to Full ZS is insufficient; comparisons with LLM fine-tuning on public data are needed. Therefore, I will maintain my score.

---

> > > ### Author Response · Authors · 2024-11-25
> > > **Our method is applicable to any task by using text-infilling & provides a significantly more efficient alternative with higher performance than fine-tuning large LLMs on public data**
> > >
> > > We thank the reviewer for taking the time to carefully read our responses and provide thoughtful feedback.
> > >
> > > >**I remain concerned that the classification task is of little value here and suggest adding more non-classification experiments.**
> > >
> > > We frame the classification task as a text-infilling task, predicting the next word in a sequence. This approach is broadly applicable, extending to tasks like text generation, as demonstrated in our experiments on MIT-D. Notably, **our method consistently outperforms all prior approaches across a variety of tasks!** To further validate our findings, we conducted additional experiments using Llama2-7b. The results, presented in the table below (and included in the main paper as Table 4), confirm that our method surpasses all baselines by a significant margin!
> > >
> > > | **Method** | **Φₜ** | **Φₛ** | **SST2** | **IMDB** | **Tweet** | **arisetv** |
> > > |------------------------------------------------|------------|-----------------|--------|--------|--------|---------|
> > > | OPT [Hong et al., 2023] | Llama2-7b | our compressed | 81.31| 67.40 | 26.90 | 82.00 |
> > > | OPT [Hong et al., 2023] | Llama2-7b | GPT-2 | 81.65| 62.93 | 41.15 | 78.26 |
> > > | Zero-Shot Transfer [Wu et al., 2023b] | Llama2-7b | our compressed | 62.38 | 70.57  | 42.80 | 58.33 |
> > > | Zero-Shot Transfer [Wu et al., 2023b] with DP | Llama2-7b | our compressed | 53.55 | 69.47 | 41.65 | 59.54 |
> > > | **POST (ours)** | Llama2-7b | our compressed | **90.14** | **86.27** | **61.70** | **86.71** |
> > > | **DP-POST (ours)** | Llama2-7b | our compressed | 89.91 | 83.26 | 59.55 | 82.60 |
> > >
> > > >**Additionally, comparing only to Full ZS is insufficient; comparisons with LLM fine-tuning on public data are needed.**
> > >
> > > The key takeaway is that fine-tuning large LLMs on public data is highly resource-intensive and provides low performance, while our approach provides a significantly more efficient alternative with higher performance for adapting large LLMs.
> > >
> > > We extended Table 3 (GPT2-XL) to include compute times, with the new results detailed in the table below and in Table 13 of the appendix. For the relatively larger SST2 dataset, prompt-tuning on the proxy model takes 310 minutes, and transferring the tuned soft prompt to the large LLM takes an additional 99 minutes, resulting in a total of 409 minutes. In contrast, directly fine-tuning a soft prompt on the large LLM requires 2660 minutes. A similar efficiency trend is observed across other datasets. For the ARISETV dataset, the process is even faster. Prompt-tuning on the smaller model takes just 23 minutes, and the transfer step adds 99 minutes, totaling 122 minutes—over an hour faster than the 184 minutes required for direct prompt-tuning on the large teacher LLM. These results clearly demonstrate the substantial time savings our method provides.
> > >
> > > | **Method** | **Runtime for arisetv (min)** | **Runtime for sst2 (min)** |
> > > |----------------------|---------------------------|-------------------------|
> > > | PT on Φ_t | 184 | 2660 |
> > > | (1) PT on Φ_s | 23 | 310 |
> > > | (2) Transfer | 99 | 99 |
> > > | **Ours total (1)+(2)** | **122** | **409** |
> > >
> > > >**Therefore, I will maintain my score.**
> > >
> > > We thank the reviewer for their thoughtful feedback, which has undoubtedly helped improve the quality of the paper. We had hoped that the new results demonstrating the applicability of our method to text generation tasks, along with its minimal computational overhead, would merit a reconsideration and potentially a higher score.

---

> ### Author Response · Authors · 2024-11-25
> **Comparison with LLM fine-tuning on public data**
>
> >**Additionally, comparing only to Full ZS is insufficient; comparisons with LLM fine-tuning on public data are needed.**
>
> We ran additional experiments to fine-tune large models on the public dataset. Our result is included as Full PT (public) in the table below. We observe that directly fine-tuning on the public dataset decreases leads to low performance on the private dataset. These results further reinforce the strengths of our proposed method, demonstrating that it not only offers significant time savings but also achieves substantially better performance compared to all baselines.
>
> | Private  | Full ZS | Full PT (private) | Full PT (public) |Compressed PT | Direct Transfer | Public | Transfer Acc |
> |----------|---------|---------|----- |----------|-----------------|--------|--------------|
> | MIT-D    | 70.84   | 92.28| 62.89   | 21.69         | 43.61           | AIE    | **75.66**       |

---

> > ### Author Response · Authors · 2024-11-25
> > **Addresed Concerns & the score**
> >
> > We thank the Reviewer for engaging with us during the discussion period. We hope that we addressed the Reviewer's concerns and would like to ask for an increase in the score.

---

> > > ### Author Response · Authors · 2024-12-01
> > > **Have concerns been addressed?**
> > >
> > > We would like to follow up on our responses, especially on the experiments for the non-classification tasks and LLM fine-tuning on public data. We show the main results in the Table below for the Reviewer's convenience. Our method was applied to the generation task (MIT-D), significantly outperforming the LLM fine-tuning on the corresponding public data. Do the results adequately address the reviewer's concerns?
> > >
> > > | Private  | Full ZS | Full PT (private) | Full PT (public) |Compressed PT | Direct Transfer | Public | Transfer Acc |
> > > |----------|---------|---------|----- |----------|-----------------|--------|--------------|
> > > | MIT-D    | 70.84   | 92.28| 62.89   | 21.69         | 43.61           | AIE    | **75.66**       |

---

### Official Review · Reviewer_y96m · 2024-11-04

**Soundness:** 2
**Presentation:** 3
**Contribution:** 2
**Rating:** 6
**Confidence:** 4

**Summary:**

Prompt tuning, the process of optimizing the prefix tokens (hard) or their embeddings (soft) to improve the utility of a downstream task has become one important way of adapting Large Language Models (LLMs). Traditional soft prompt tuning paradigms require backpropagation through the full model, which is computationally expensive to perform on-premise, or handing the private data to a central LLM provider, which could lead to privacy concerns. To reduce the computational overhead of local soft prompt tuning while preserving privacy, the authors propose Privacy Of Soft-prompt Transfer (POST), which performs soft prompt tuning on a small local LLM distilled from a large LLM and transfers the soft prompt to the large model with th help of public data. In addition, the proposed framework can also be coupled with differentially private prompt tuning techniques to achieve privacy guarantees. To validate their proposed method, the authors evaluate the transferability of soft prompts finetuned on small models distilled from Roberta-base, GPT2-XL, and Llama2-7b for classification tasks.

**Strengths:**

The setting of the paper is well-motivated. The introduction sections and the figures clearly explain the setting and the proposed methods.

**Weaknesses:**

1. While the paper presents an appealing middle ground by performing prompt finetuning on a distilled proxy model, the evaluations are performed with a fixed set of proxy models. As briefly mentioned by the authors, in the case of the centralized LLM provider, the utility of the distilled model should ideally be not strong enough to replace the original model but is still good enough to serve as a proxy for prompt finetuning. However, the authors do not further explore the trade-off between the quality of the proxy model and the effectiveness of the proposed pipeline.
2. The evaluation of the proposed method mainly focuses on classification tasks, which have limited practicality given that modern autoregressive LLMs can perform open-ended generation tasks. While I recognize that prior works also mainly focus on classification tasks, I encourage the authors to discuss the applicability of the proposed method on open-ended tasks, or even better, demonstrate such ability through follow-up experiments.

**Questions:**

1. I appreciate the authors discussing the runtime of POST v.s. Full PT in Table 3, but I have some questions regarding the costs of the distillation step. While the distillation costs of a large LLM can be amortized as we finetune more soft prompts with the distilled LLM, could the authors quantitatively compare the computational overhead of finetuning one soft prompt on the large LLM to that of distilling the large LLM and finetuning one soft prompt on the distilled LLM?
2. The authors use a fixed set of distilled models throughout the paper. How does the quality of the distilled model influence the transferability?
3. Could other model compression techniques like pruning and quantization be used in this framework instead of model distillation?

---

> ### Author Response · Authors · 2024-11-24
>
> >**However, the authors do not further explore the trade-off between the quality of the proxy model and the effectiveness of the proposed pipeline. The authors use a fixed set of distilled models throughout the paper. How does the quality of the distilled model influence the transferability?**
>
> We conducted additional experiments on the trade-off of the quality of the proxy model and the transferred performance. We measure the quality of the proxy model by 1) it’s downstream task performance and 2) the training loss value at a given checkpoint. The new results are shown in Figure 6 in the appendix of the revised submission. We distilled 4 different sizes of roberta-base model and used the checkpoints at 50,000, 100,000, 200,000, and 500,000 steps to perform the transfer. We also include the results as tables below ( please refer to Figure 6 in the submitted paper for better visualization of the trends).
>
> We report Transferred Acc (Proxy model Acc). Each value represents the transferred accuracy (the Transfer Acc. represented by all values not in parentheses) and the accuracy on the proxy model (Proxy model Acc - for all values in the parentheses).
>
> | checkpoint_model         | 50000         | 100000        | 200000        | 500000        |
> |:-------------------------|:--------------|:--------------|:--------------|:--------------|
> | distilled_roberta-1layer | 51.09 (15.61) | 48.19 (19.02) | 46.86 (21.39) | 49.28 (25.66) |
> | distilled_roberta-2layer | 73.19 (70.18) | 75.24 (71.71) | 81.28 (70.56) | 79.47 (69.21) |
> | distilled_roberta-3layer | 85.27 (75.41) | 84.78 (75.59) | 85.75 (75.77) | 87.08 (74.09) |
> | distilled_roberta-4layer | 81.16 (77.54) | 82.97 (78.61) | 84.78 (78.05) | 86.35 (78.63) |
>
> The results show that, in general, the higher quality of the proxy model results in the higher the transferred performance (and the trend is strengthened with more layers in the proxy model).
>
> >**While I recognize that prior works also mainly focus on classification tasks, I encourage the authors to discuss the applicability of the proposed method on open-ended tasks, or even better, demonstrate such ability through follow-up experiments.**
>
> We thank the Reviewer for the suggestion. Although in the original submission, we presented results on classification tasks, we always format the classification task as a text-infiling task which generates the next word prediction. This makes our approach inherently suitable to address other tasks as well. Thus, **we performed new experiments on open-ended Q&A tasks**. We evaluated our method’s effectiveness on the MIT-D movie dataset consisting of 1561 train and 415 test samples. The task is to extract a movie’s director from a given movie description. Instead of generating a single token as in the classification tasks, this task requires generating multiple tokens with varying lengths. The result is shown below as well as in Table 12 in the appendix. Our method’s performance (Transfer Acc) is higher than Full ZS (Zero-Shot) and Compressed PT (Prompt Tuning), highlighting our method’s applicability to open-ended tasks.
>
> | Private  | Full ZS | Full PT | Compressed PT | Direct Transfer | Public dataset | Transfer Acc |
> |----------|---------|---------|---------------|-----------------|--------|--------------|
> | MIT-D    | 70.84   | 92.28   | 21.69         | 43.61           | AIE    | **75.66**  |

---

> ### Author Response · Authors · 2024-11-24
>
> >**could the authors quantitatively compare the computational overhead of finetuning one soft prompt on the large LLM to that of distilling the large LLM and finetuning one soft prompt on the distilled LLM?**
>
> We extended our Table 3 (GPT2-XL) with the compute times according to the reviewer’s suggestions, the new results are listed in Table 13 in the appendix.
> Our distillation takes 1203 min under this setting. For a relatively larger SST2 dataset, the prompt-tuning on the proxy model takes 310 min and the transfer takes 99 min. Overall, this gives **us 1612 min** for distilling the large LLM, finetuning one soft prompt on the distilled LLM, and transferring the soft prompt to the large LLM. In contrast, directly fine-tuning one soft prompt on the large LLM takes **2660 min**. Thus, in this case, **our method does not incur any computational overhead but is actually faster than the direct prompt tuning on the large teacher LLM**. On the other hand, for a very small arisetv dataset, this comparison is 1405 min vs. 184 min. Although we are slower under this scenario, we need to emphasize that the distillation is only done once and can be used for other tasks. Note that the prompt tuning takes only 23 min on the small model and the transfer lasts 99 min, hence overall only 122 min which is still faster (by more than 1h) than the 184 min prompt tuning on the large teacher LLM.
>
> These results show that for standard-large datasets, our method is already faster in comparison to tuning *one single prompt on the large model*. For small datasets, the distillation time amortizes after training a few soft prompts and gives our method an advantage over the direct prompt tuning on large LLMs.
>
>
> | **Method** | **Runtime for arisetv (min)** | **Runtime for sst2 (min)** |
> |----------------------|---------------------------|-------------------------|
> | PT on Φ_t | 184 | 2660 |
> | (1) PT on Φ_s | 23 | 310 |
> | (2) Transfer | 99 | 99 |
> | **Ours total (1)+(2)** | **122** | **409** |
> | (3) KD | 1203| 1203|
> |**Ours total (1)+(2)+(3)** |1405 |1612 |
>
>
> >**Could other model compression techniques like pruning and quantization be used in this framework instead of model distillation**
>
> We thank the reviewer for their insightful question on other compression techniques.
>
> **Quantization** reduces the precision of model parameters to lower bit-widths, effectively decreasing model size. However, since our framework involves transmitting the proxy model to the user, quantization *primarily reduces storage requirements without adequately protecting the intellectual property inherent in the original large model*. Additionally, quantized models often require specialized hardware or software support to maintain performance, which may not be available to all users.
>
> **Pruning** involves removing less significant weights or neurons from the model. *Unstructured pruning*, which eliminates individual weights, *typically does not lead to practical speedups or reduced resource consumption, making it less applicable to our approach*. Structured pruning, on the other hand, removes entire neurons, filters, or layers, resulting in a smaller and more efficient model. However, *many structured pruning methods are task-specific and require fine-tuning on specific tasks to regain performance.*

---

> > ### Comment · Reviewer_y96m · 2024-11-24
> > **Thanks for the reply**
> >
> > I thank the authors for their detailed response. I will keep my current score as is.

---

> > > ### Author Response · Authors · 2024-11-24
> > > **Thank you for your response**
> > >
> > > We thank the reviewer for their thoughtful feedback, which has undoubtedly helped improve the quality of the paper. We had hoped that the new results demonstrating the applicability of our method to text generation tasks, along with its minimal computational overhead, would merit a reconsideration and potentially a higher score.

---

### Author Response · Authors · 2024-11-24
**General Response**

We sincerely thank the reviewers for their thoughtful feedback and for recognizing the strengths of our work. We are delighted that our paper addresses an important and well-motivated problem in the field (Reviewer y96m). Our work is further acknowledged for proposing a novel framework for privacy-preserving soft prompt transfer, effectively tackling challenges in privacy, efficiency, and computational cost while achieving competitive performance (Reviewers mvG3 and QrLz). Additionally, the clarity in explaining related work and key concepts, as well as the detailed description of the proposed algorithm, has been positively highlighted (Reviewer dYht). We hope that our method serves as a step forward in encouraging providers of closed LLMs, such as OpenAI and Google, to enable soft-prompt tuning.

Beyond the classification tasks, we present a new experiment on open-end generation tasks which are not shown in the previous work (Hong et al., 2023 and Wu et al., 2023b). This task requires generating multiple tokens with varying lengths. Our results in the following table show that our method’s performance (Transfer Acc) is higher than Full ZS (Zero-Shot) and Compressed PT (Prompt Tuning), highlighting our method’s applicability to open-ended tasks.

| Private  | Full ZS | Full PT | Compressed PT | Direct Transfer | Public dataset | Transfer Acc |
|----------|---------|---------|---------------|-----------------|--------|--------------|
| MIT-D    | 70.84   | 92.28   | 21.69         | 43.61           | AIE    | **75.66**  |

---

### Meta-Review · Area_Chair_a1Dd · 2024-12-20

**Metareview:**

Prompt tuning is an effective way to adapt LLM by learning and specifying prefix tokens. This paper proposes a privacy-preserving soft prompt tuning without accessing the LLMs. Instead, the service model provides a small and distilled model to the user, who will then perform soft prompt tuning at local. The soft prompt will then be transferred back with the help of public datasets.

Reviewers had questions on the motivations for the specific setting studied in the paper. For proprietary service providers, distilling smaller models and sending them to users seem impractical, while it’s unclear if the privacy concern remains equally important when using open-source models. In addition, reviewers raised questions about lacking systematic guidelines for selecting public datasets, missing comprehensive ablation studies for hyperparameters etc.

**Additional Comments On Reviewer Discussion:**

After rebuttals, the primary concerns remained with the reviewers.

---

### Decision · Program_Chairs · 2025-01-22

Reject